# Parenting interventions to promote early child development in the first three years of life: A global systematic review and meta-analysis

**Joshua Jeong** [ID]\*, **Emily E. Franchett** [ID], **Clariana V. Ramos de Oliveira** [ID], **Karima Rehmani** [ID], **Aisha K. Yousafzai** [ID]

Department of Global Health and Population, Harvard T.H. Chan School of Public Health, Boston, Massachusetts, United States of America

\* joshua.jeong@hsph.harvard.edu

## Abstract

### Background

Parents are the primary caregivers of young children. Responsive parent–child relationships and parental support for learning during the earliest years of life are crucial for promoting early child development (ECD). We conducted a global systematic review and meta-analysis to evaluate the effectiveness of parenting interventions on ECD and parenting outcomes.

### Methods and findings

We searched MEDLINE, Embase, PsycINFO, CINAHL, Web of Science, and Global Health Library for peer-reviewed, published articles from database inception until November 15, 2020. We included randomized controlled trials (RCTs) of parenting interventions delivered during the first 3 years of life that evaluated at least 1 ECD outcome. At least 2 reviewers independently screened, extracted data, and assessed study quality from eligible studies. ECD outcomes included cognitive, language, motor, and socioemotional development, behavior problems, and attachment. Parenting outcomes included parenting knowledge, parenting practices, parent–child interactions, and parental depressive symptoms. We calculated intervention effect sizes as the standardized mean difference (SMD) and estimated pooled effect sizes for each outcome separately using robust variance estimation meta-analytic approaches. We used random-effects meta-regression models to assess potential effect modification by country-income level, child age, intervention content, duration, delivery, setting, and study quality. This review was registered with PROSPERO (CRD42018092458 and CRD42018092461). Of the 11,920 articles identified, we included 111 articles representing 102 unique RCTs. Pooled effect sizes indicated positive benefits of parenting interventions on child cognitive development (SMD = 0.32, 95% CI [confidence interval]: 0.23, 0.40, *P* < 0.001), language development (SMD = 0.28, 95% CI: 0.18 to 0.37, *P* < 0.001), motor development (SMD = 0.24, 95% CI: 0.15 to 0.32, *P* < 0.001), socioemotional development (SMD = 0.19, 95% CI: 0.10 to 0.28, *P* < 0.001), and attachment (SMD =

**Data Availability Statement:** All relevant data are within the manuscript and its Supporting information files.

**Funding:** This work was funded by the World Health Organization (grant # APW-201931623 awarded to AKY). The funder had no role in study design, data extraction and analysis, decision to publish, or preparation of the manuscript.

**Competing interests:** The authors have declared that no competing interests exist.

**Abbreviations:** BSID, Bayley Scales of Infant and Toddler Development; CI, confidence interval; ECD, early child development; HICs, high-income countries; LMICs, low- and middle-income countries; PRISMA, Preferred Reporting Items for Systematic Reviews and Meta-Analysis; RCTs, randomized controlled trials; SMD, standardized mean difference.

0.29, 95% CI: 0.18 to 0.40, $P < 0.001$) and reductions in behavior problems (SMD = −0.13, 95% CI: −0.18 to −0.08, $P < 0.001$). Positive benefits were also found on parenting knowledge (SMD = 0.56, 95% CI: 0.33 to 0.79, $P < 0.001$), parenting practices (SMD = 0.33, 95% CI: 0.22 to 0.44, $P < 0.001$), and parent–child interactions (SMD = 0.39, 95% CI: 0.24 to 0.53, $P < 0.001$). However, there was no significant reduction in parental depressive symptoms (SMD = −0.07, 95% CI: −0.16 to 0.02, $P = 0.08$). Subgroup analyses revealed significantly greater effects on child cognitive, language, and motor development, and parenting practices in low- and middle-income countries compared to high-income countries; and significantly greater effects on child cognitive development, parenting knowledge, parenting practices, and parent–child interactions for programs that focused on responsive caregiving compared to those that did not. On the other hand, there was no clear evidence of effect modification by child age, intervention duration, delivery, setting, or study risk of bias. Study limitations include considerable unexplained heterogeneity, inadequate reporting of intervention content and implementation, and varying quality of evidence in terms of the conduct of trials and robustness of outcome measures used across studies.

## Conclusions

Parenting interventions for children during the first 3 years of life are effective for improving ECD outcomes and enhancing parenting outcomes across low-, middle-, and high-income countries. Increasing implementation of effective and high-quality parenting interventions is needed globally and at scale in order to support parents and enable young children to achieve their full developmental potential.

## Author summary

### Why was this study done?

- Parenting interventions have been underscored as a key strategy for improving early child development (ECD) outcomes.

- Although there are several existing reviews regarding the effectiveness of parenting interventions for improving ECD outcomes, prior reviews have focused narrowly on select types of parenting interventions, evaluated impacts on single ECD domain outcomes, included studies in either only high-income countries (HICs) or low- and middle-income countries (LMICs), and have not adequately explored treatment heterogeneity and potential moderators.

### What did the researchers do and find?

- We conducted a systematic review and meta-analysis of 102 randomized controlled trials of parenting interventions for children during the first 3 years of life that were implemented across a total of 33 countries.

- We found that parenting interventions improved early child cognitive, language, motor, socioemotional development, and attachment and reduced behavior problems. Parenting interventions additionally improved parenting knowledge, parenting practices, and parent–child interactions. However, they did not significantly reduce parental depressive symptoms.

- We found that parenting interventions had significantly greater effects on child cognitive, language, and motor development and parenting practices in LMICs than HICs (e.g., effect on cognitive development was 3 times greater in LMICs versus HICs). Parenting interventions that included content on responsive caregiving had significantly greater effects on child cognitive development, parenting knowledge, parenting practices, and parent–child interactions than interventions that did not include content on responsive caregiving (e.g., effect on parenting practices was nearly 4 times greater for interventions with responsive caregiving content versus those without responsive caregiving content).

- We uncovered substantial variation in program content and implementation characteristics across studies and considerable heterogeneity in pooled effect size estimates across nearly all evaluated outcomes.

**What do these findings mean?**

- To the best of our knowledge, this is the largest and most comprehensive global systematic review and meta-analysis that demonstrates the effectiveness of parenting interventions during the first 3 years of life on a wide range of ECD and parent-level outcomes.

- Parenting programs are needed globally to enhance parent–child relationships and promote the healthy development of children during the earliest years of life.

- Future research should unpack the observed variability in program components and implementation features used across parenting interventions and examine their associations with outcomes to inform improved delivery, effectiveness, and scale of parenting interventions for ECD.

## Introduction

Globally, an estimated 250 million children under aged 5 years (43%) are at risk of not achieving their developmental potential in the earliest years of life due to a host of nutritional, health, and psychosocial risks [1–3]. Promotive interventions are particularly important in the first 3 years of life when the developing brain is most sensitive to experiences and the environment [4]. While various types of interventions including nutrition and health can support healthy development, recent evidence has revealed that parenting interventions that include components to directly enhance early child learning or strengthen parent–child relationships are more effective for improving early cognitive, language, motor, and socioemotional development [5–7].

Parenting interventions are social and behavioral programs intended to improve caregivers' knowledge, attitudes, practices, and skills in order to promote optimal early child development

(ECD) [8]. Parenting interventions can encompass a range of interventions targeting various risks, behaviors, or aspects of parent–child relationships, including interventions that focus on stimulation [6], shared book reading [9], attachment and parental sensitivity [10,11], behavior management [12,13], positive discipline and maltreatment prevention [14,15], and parental mental health [16,17]. These different types of parenting interventions have consistently shown benefits across a wide age range of children, and especially during early childhood [5,18].

Despite a significant rise over the past decade in the implementation and quality of parenting interventions globally, there has not been a recent review comprehensively synthesizing the latest state of the literature. One of the most recent and relevant reviews on parenting interventions during early childhood was published by Britto and colleagues [5] as part of *The Lancet* 2017 series on ECD. This review found significant small-to-medium sized benefits of parenting interventions on child cognitive (standardized mean difference (SMD) = 0.36, $n$ = 19 studies), motor (SMD = 0.13, $n$ = 9), and socioemotional development (SMD = 0.35, $n$ = 13). However, this evidence synthesis was based on a review of 3 selected reviews published in 2014 [19] and 2015 [6,20] that focused only on ECD but not parenting outcomes and was limited to studies in low- and middle-income countries (LMICs). Another relevant review by Jeong, Pitchik, and Yousafzai [21] investigated the effectiveness of stimulation interventions during the first 3 years of life on parent-level outcomes and documented significant medium-to-large benefits on maternal knowledge (SMD = 0.91, $n$ = 6), parenting practices (SMD = 0.57, $n$ = 10), and mother–child interactions (SMD = 0.44, $n$ = 3). No significant effect was observed for reducing maternal depressive symptoms (SMD = −0.10, $n$ = 9). Yet this prior review focused specifically on stimulation interventions, examined effects on parenting but not ECD outcomes, and was also limited to LMICs.

While these and other reviews have underscored positive effects of particular types of parenting interventions on ECD outcomes [22,23], an updated review is overdue and needed to improve our current understanding of the effectiveness of parenting interventions during early childhood and generalizability of the evidence base. Additionally, little attention has been given previously to the role of intervention implementation characteristics as potential moderators of program effects on child or caregiving outcomes. Considering the diversity of contexts and populations in which parenting programs are implemented, there is a need to investigate whether certain implementation factors are associated with differential program effectiveness.

Therefore, the present study aimed to provide a comprehensive global review of the effectiveness of parenting interventions delivered during the first 3 years of life on both ECD and parent-level outcomes. This study updates, synthesizes, and expands on previous reviews by (1) adopting an inclusive definition of parenting interventions as those that aim to improve caregiver interactions, behaviors, knowledge, beliefs, attitudes, or practices with their children in order to improve ECD; (2) focusing on interventions targeting caregivers and children during the first 3 years of life; (3) considering a broad set of both ECD and caregiver outcomes; and (4) including studies across low-, middle-, and high-income countries. The primary aim was to quantify the pooled effectiveness of parenting interventions delivered during the first 3 years of life on ECD and parenting-related outcomes. The secondary aim was to explore whether effects on ECD and parenting outcomes differed by country income level, child age, intervention content, duration, delivery, setting, and study quality.

## Methods

### Search strategy and selection criteria

This systematic review and meta-analysis was conducted and reported using the Preferred Reporting Items for Systematic Reviews and Meta-Analysis (PRISMA) guidelines [24] (S1

PRISMA Checklist). Six electronic bibliographic databases (MEDLINE, Embase, PsycINFO, CINAHL, Web of Science, and Global Health Library) were searched for articles published from database inception until May 15, 2019. The search was subsequently updated during the peer-review process to include articles published between May 16, 2019 to November 15, 2020. The search strategy (S1 Text) was informed by search terms and keywords used in prior systematic reviews related to parenting interventions [5,6,25] and through consultations with a research librarian. Reference lists of relevant studies and reviews were also scanned for any additional studies that may have been missed.

Full-text, peer-reviewed articles were included if they met the following criteria: (1) parenting interventions that aimed to improve interactions, behaviors, knowledge, beliefs, attitudes or practices of parents with their children in order to improve ECD; (2) evaluated using a randomized controlled study design; (3) targeted children and their parents during early childhood (pregnancy through the first 3 years of life); and (4) measured at least 1 ECD outcome after the completion of the intervention (or shortly thereafter). In this study, parents are broadly defined as the legal guardian, biological parent, or adult caregiver responsible for the well-being of the child [20]. Studies were excluded if they met any of the following criteria: (1) not a relevant parenting intervention focused on promoting ECD; (2) nonrandomized study design (e.g., quasi-experimental studies); (3) targeted a population of children who were, on average, older than 36 months; (4) targeted a population of children or parents who had a diagnosed illness or disability; and/or (5) did not measure at least 1 ECD outcome.

Titles and abstracts of all identified articles were independently screened by at least 2 reviewers (JJ, EF, and a team of graduate research assistants). If the title and abstract screening was insufficient for determining eligibility, the full-text article was reviewed for eligibility criteria. Any discrepancies between the 2 reviewers regarding the eligibility of a study were resolved through discussions among the reviewers, with input from another author (AKY) as needed until consensus was reached.

For studies meeting the eligibility criteria, full-text articles were reviewed. Data were extracted by JJ, EF, and a team of trained graduate research assistants using a prepiloted extraction form. JJ and EF trained the graduate research assistants over the course of a 3-week training period on how to use the data extraction sheet based on pilot exercises of relevant articles. Throughout the entire data extraction process, each data extraction completed by the research assistants was independently extracted by a second reviewer (JJ or EF), and any issues were discussed and resolved during the weekly in-person team meetings with the research assistants. For additional quality assurance, 35% of assigned articles were randomly selected for independent data extraction by a third reviewer (JJ or EF). Any discrepancies between the master reviewers (JJ and EF) regarding the extracted data were resolved through discussions and consensus together with an additional reviewer (AKY).

## Outcomes

The primary outcomes were 6 domains of ECD: cognitive, language, motor, and socioemotional development, behavior problems, and attachment. These ECD outcome domains have been prioritized in previous work, commonly measured across global contexts, and are each unique aspects of child development during the first 3 years of life [26,27]. Moreover, each of these ECD outcome domains has shown predictive validity with later achievement, function, performance, or outcomes [27]. Measures of developmental disabilities (e.g., autism or ADHD) did not qualify as ECD outcomes. The secondary outcomes were 4 measures of parenting: parenting knowledge, parenting practices, parent–child interactions, and parental depressive symptoms.

## Data analysis

Effect sizes on ECD and parenting outcomes were calculated as SMD between the intervention and comparison arms (e.g., no intervention, nonparenting intervention, standard of care), with respect to change in mean values from baseline to endline (when baseline values were reported) after standardization by their pooled standard deviation (SD). In multiarm studies, the comparison group was typically the intervention arm without any parenting components and/or the standard of care. For studies that reported outcomes from multiple follow-up waves of assessments, outcome measurements from the time point closest to the completion of the intervention were used for the analyses. The magnitude of effect sizes was interpreted in the context of public health, pediatric, and early education interventions and the practical significance with respect to the outcomes [28].

Each of the primary and secondary outcomes was examined in separate models. We used a robust variance estimation meta-analysis model [29] to account for multiple measurements per trial per outcome domain. This approach allows for the inclusion of any number of dependent effect size estimates from a given study to contribute to the pooled effect size estimate for an outcome, resulting in increased power and more precise estimates. For example, this approach was used to handle the separately reported effect sizes for receptive language and expressive language in a given study for overall language development or both internalizing and externalizing behaviors for overall child behavior problems. Pooled effect size estimates were based on random effects models. $P$ values $< 0.05$ denoted statistical significance. Heterogeneity of the pooled effect size was assessed using the $I^2$ statistic and the Cochran $Q$ test and its $P$ value.

Moderator analyses were conducted on all outcomes using random effects meta-regression models to explore potential sources of heterogeneity in the average effect by the following prespecified sample, intervention, and study characteristics: country-income level (LMICs or high-income countries (HICs)), average child age at baseline in months, whether the parenting intervention included a component to enhance responsive caregiving (see S1 Table for classification of responsive caregiving interventions) [25,30], intervention duration in months, delivery modality (individual, group, or both), setting (home, community, clinic, or combination), and study quality score. Study quality was assessed using the Cochrane Collaboration Risk of Bias Assessment Tool for randomized controlled trials (RCTs) [31]. Studies were rated as low risk (0 points), unclear risk (1 point), and high risk (2 points) for a total risk of bias score ranging from 0 to 16 [22,32]. For the continuous moderator variables, we generated a binary variable for child age at baseline and intervention duration ($<$ or $\geq$12 months) and a binary study quality variable based on a mean split of the total risk of bias score (lower or higher risk of bias) and compared effect estimates according to these subgroups. Statistical significance for the moderator variables of country-income level, intervention content, delivery, and setting was determined from the $Q_{between}$ test statistic which was evaluated on the $\chi^2$ distribution. For the remaining moderator variables that were originally continuous measures (i.e., child age, intervention duration, and study quality score), statistical significance was determined from meta-regressions using the continuous measure (e.g., child age in months).

Publication bias was examined for each outcome analysis using the Egger's regression test to test the null hypothesis of small-study bias (i.e., the extent to which less precise or smaller sample studies yield greater effects than larger sample studies) [33]. All analyses were conducted using Stata version 16. This review was preregistered with PROSPERO (CRD42018092458 and CRD42018092461). See S1 Protocol for the final review protocol used for the present study and notes regarding any changes made from the initial prospectively registered protocol.

## Results

### Study selection and inclusion

A total of 11,906 unique records were identified from electronic databases. Fourteen additional records were identified from article references and subject matter expertise. A total of 11,580 records were excluded based on title and abstract screening. The full texts of the remaining 340 articles were reviewed, and 229 articles were excluded for not meeting inclusion criteria. The final set comprised of 111 articles representing 102 unique RCTs (Fig 1).

### Sample, intervention, and study characteristics

Table 1 presents the characteristics of the 102 RCTs of parenting interventions. Trials were implemented across a total of 33 countries. The majority of trials were conducted in HICs (61 trials in 14 countries), compared to LMICs (41 trials in 19 countries). The most represented countries were the United States of America (40 trials), Bangladesh (8 trials), Jamaica (6 trials),

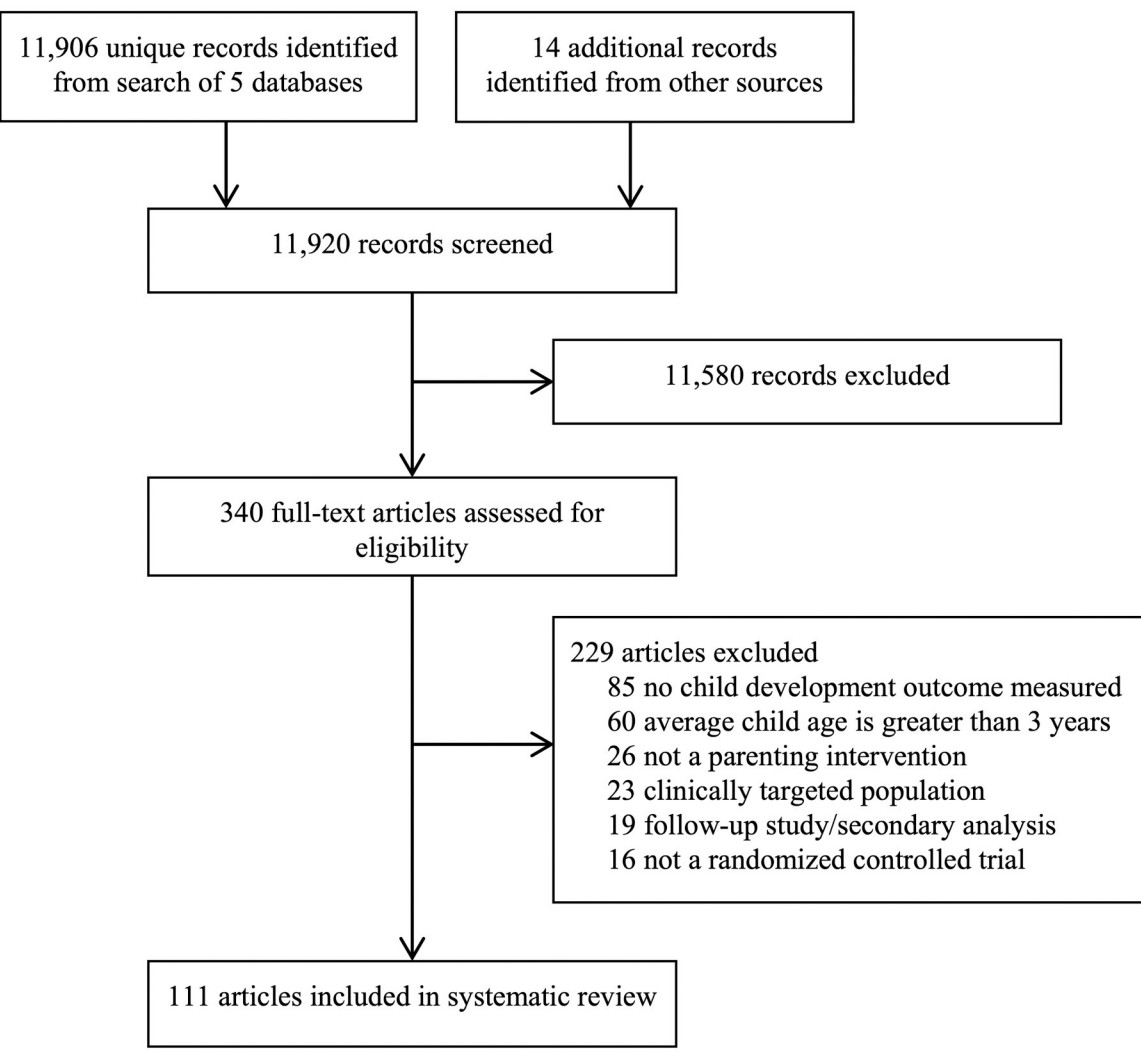

**Fig 1. Preferred Reporting Items for Systematic Reviews and Meta-analyses (PRISMA) flow diagram of search results and included studies.**

**Table 1. Characteristics of included parenting interventions.**

| Author, Year Country | Intervention Description | Intervention Includes Responsive Caregiving | Average Child Age at Baseline | Average Program Duration | Study Design (Unit of Randomization) | Delivery | Setting | Analytic Sample | Early child development outcomes | | | | | | Parenting outcomes | | | Depression |
|---|---|---|---|---|---|---|---|---|---|---|---|---|---|---|---|---|---|---|
| | | | | | | | | | Cognitive | Language | Motor | Socio-emotional | Behavior problems | Attachment | Knowledge | Practices | Parent-child interactions | |
| Abessa et al., 2019 [74] Ethiopia | A family-based psychomotor/psychosocial stimulation intervention to provide treatment for children with severe acute malnutrition. Trained intervention nurses delivered hospital-based individual and group sessions to provide stimulation for children through play and to coach caregivers on how to stimulate their children. After being discharged, families received play materials and 3 follow-up home visits from play leaders who provided continued coaching. | No | 27.4 mo | 6.5 mo | Two-arm RCT (Individual) | Individual, Group | Hospital, Home | INT: 113 CTRL: 98 | | X | X | X | | | | | | |
| Abimpaye et al., 2019 [39] Rwanda | First Steps, or "Intera za Mbere", is a parenting education program utilizing a curriculum that focuses on promoting responsive caring and bonding, providing stimulation through playful learning, supporting child health, and promoting emergent literacy in the home. Both the "light touch" and "full" intervention groups received weekly parenting education sessions facilitated by radio, supported by a local volunteer. Additional elements in the "full intervention" group included more intensive training for the local volunteer, book gifting to participating families, and support from a facilitator to guide group sessions and conduct home visits. | Yes | 6-24 mo | 4.25 mo | Three-arm Cluster-RCT (Sector) | Individual, Group | Home, Community | INT: 486 CTRL: 479 | X | X | X | X | | | | X | | |
| Aboud and Akhter, 2011 [75] Bangladesh | An intervention delivering informational sessions on health, nutrition, and child development to mothers. Mothers in the Responsive Feeding and Stimulation (RFS) intervention group received additional sessions delivering messages on RFS supported through discussion, demonstrations, practice and coaching. A second intervention group, RFS+, received the RFS intervention plus Sprinkles micronutrient powder for 6 months. | Yes | 14.2 mo | 7 mo | Three-arm Cluster RCT (Village) | Group | Community | INT: 188 CTRL: 105 | | X | | | | | | X | | |
| Aboud et al., 2013 [76] Bangladesh | A parenting intervention providing mothers with messages on health and hygiene, nutrition, responsive feeding, communication, and play through a combination of group meetings and home visits. Intervention delivery agents supplied mothers with information, demonstrations, and practice supported by coaching. | Yes | 8.8 mo | 10 mo | Two-arm Cluster RCT (Union) | Individual, Group | Home, Community, Clinic | INT: 221 CTRL: 226 | X | X | | | | | X | X | | X |
| Alvarenga et al., 2019 [77] Brazil | The Maternal Sensitivity Program is a home visiting intervention targeting mothers and infants from low-income families. It uses modeling and short video feedback of mother–child interactions to enhance maternal sensitivity and child development. | Yes | 3 mo | 8 mo | Two-arm RCT (Mother–infant dyad) | Individual | Home | INT: 22 CTRL: 22 | | | | | X | | | | X | |
| Andrew et al., 2020 [78] India | This psychosocial stimulation intervention comprised weekly 1-hour home visits delivered by local home visitors for 18 months. The intervention followed a structured curriculum of developmentally appropriate activities using low-cost homemade toys and picture books, based on the Reach Up and Learn model and adapted to the local context. Home visitors demonstrated activities, provided caregivers opportunities to practice, and encouraged caregivers to respond to their children's actions and vocalizations. | Yes | 14.9 mo | 18 mo | Two-arm Cluster RCT (Community) | Individual | Home | INT: 191 CTRL: 187 | X | X | X | | | | X | X | | X |
| Anzman-Frasca et al., 2018 [79] USA | Intervention in which nurses delivered responsive parenting guidance through home visits and 1 center-based visit to promote sleep, feeding, soothing, interactive play, and supporting parents' contingent, appropriate responses to infant signals. | Yes | 0.5 mo | 12 mo | Two-arm RCT (Mother–infant dyad) | Individual | Home, Clinic | INT: 123 CTRL: 117 | | | | | X | | | | | |

(Continued)

**Table 1.** (Continued)

| Author, Year Country | Intervention Description | Intervention Includes Responsive Caregiving | Average Child Age at Baseline | Average Program Duration | Study Design (Unit of Randomization) | Delivery | Setting | Analytic Sample | Early child development outcomes | | | | | | Parenting outcomes | | | |
|---|---|---|---|---|---|---|---|---|---|---|---|---|---|---|---|---|---|---|
| | | | | | | | | | Cognitive | Language | Motor | Socio-emotional | Behavior problems | Attachment | Knowledge | Practices | Parent-child interactions | Depression |
| Ara et al., 2019 [40] Bangladesh | This study combined an infant feeding peer-counseling program with psychosocial stimulation to improve children's development, growth, and child-feeding practices, as compared to receipt of usual health messages. The counseling took place at home and included the mothers and key family members, and group sessions were held biweekly. Intervention participants also received a feeding bowl and spoon, handwashing solution, and homemade toys. | No | Antenatal | 12 mo | Two-arm Cluster RCT (City-block clusters) | Individual, Group | Home, Community | INT: 100 CTRL: 91 | X | X | X | X | | | | | | |
| Attanasio et al., 2014 [80] Colombia | An intervention containing a psychosocial stimulation component in which mother leaders demonstrated play activities using locally made, low-cost materials with mothers during weekly home visits to improve the quality of mother–child interactions and encourage developmentally appropriate learning activities. The nutrition component of the intervention entailed distribution of Sprinkles micronutrient powder. | Yes | 18.2 mo | 18 mo | 2 × 2 Factorial Cluster RCT (Municipality) | Individual | Home | INT: 637 CTRL: 626 | X | X | X | | | | | X | | X |
| Barlow et al., 2007 [81] UK | A weekly home visiting program for vulnerable families, beginning during pregnancy, designed to support parents and promote parent–child interaction using the Family Partnership Model. | Yes | Antenatal | 18 mo | Two-arm RCT (Mother) | Individual | Home | INT: 62 CTRL: 59 | X | | | X | | | | X | X | X |
| Barrera et al., 1986 [82] Canada | An intervention targeting preterm, low birth weight infants, which focused on improving the quality of parent–child interactions by enhancing parents' observational skills as well as their sensitivity and mutual responsivity. A second intervention group received support from intervention facilitators to enhance general understanding and awareness of child development milestones. | Yes | 0 mo | 12 mo | Three-arm RCT (Infant) | Individual | Home | INT: 38 CTRL: 21 | X | X | X | | | | | X | X | |
| Brooks-Gunn et al., 1992 [83]; Infant Health and Development Program, 1990 [84] USA | The Infant Health and Development Program is a program for low birth weight preterm infants. Services included weekly home visits in year 1 and biweekly visits in years 2 and 3 to provide information on child health and development, promote age-appropriate activities, and support problem-solving. Parents also had access to center-based childcare services and parenting group meetings. | No | 0 mo | 36 mo | Two-arm RCT (Infant) | Individual, Group | Home, Child development center, Community | INT: 377 CTRL: 608 | X | | X | | X | | | | | |
| Caughy et al., 2004 [85] USA | The Healthy Steps for Young Children Program is a universal program targeting caregivers with children aged 0–3 years. Offers a package of services to supplement standard pediatric care, including introduction of a Healthy Steps Specialist to deliver enhanced well-child visits with a pediatrician and home visits, with links to parenting group meetings and community resources. | No | 4.3 mo | 36 mo | Two-arm RCT (Family) | Individual, Group | Home, Community, Clinic, Virtual | INT: 109 CTRL: 70 | | | | | X | X | | X | X | |
| Chang et al., 2015 [86] Jamaica, Antigua, St. Lucia | An intervention integrating a parent training package into routine primary health services for children. Modules were delivered by community health workers through group sessions and individual visits with a nurse with mother–child dyads and focused on responsive stimulation and care, providing demonstrations and opportunities to practice parent–child activities and interactions using age-appropriate books and materials. | Yes | 1.5 mo | 16.5 mo | Two-arm Cluster RCT (Public health center) | Individual, Group | Clinic | INT: 210 CTRL: 216 | X | X | X | | | | X | X | | X |

(Continued)

**Table 1.** (Continued)

| Author, Year Country | Intervention Description | Intervention Includes Responsive Caregiving | Average Child Age at Baseline | Average Program Duration | Study Design (Unit of Randomization) | Delivery | Setting | Analytic Sample | Early child development outcomes | | | | | | Parenting outcomes | | | |
|---|---|---|---|---|---|---|---|---|---|---|---|---|---|---|---|---|---|---|
| | | | | | | | | | Cognitive | Language | Motor | Socio-emotional | Behavior problems | Attachment | Knowledge | Practices | Parent-child interactions | Depression |
| Cheng et al., 2007 [87] Japan | A home-based intervention to support mother–child relationships by promoting maternal sensitivity, maternal confidence in caretaking, and quality of mother–child interactions. Intervention group mothers received 5 monthly home visits from nurse home visitors, incorporating observations of mother–child interactions and strengths-based coaching. | Yes | 4 mo | 5 mo | Two-arm RCT (Mother–infant dyad) | Individual | Home | INT: 46 CTRL: 39 | | | | | X | | | | | |
| Constantino et al., 2001 [88] USA | This intervention enrolled disadvantaged urban families with infants aged 3–18 months to deliver a 10-session curriculum through a parenting group to promote secure attachment, enhance parent–child relationships, and to encourage enrollment in home visitation services. | Yes | 8.8 mo | 2.5 mo | Two-arm RCT (Family) | Group | Community | INT: 24 CTRL: 21 | | | | | X | | | | X | |
| Cooper et al., 2009 [89]; Murray et al., 2016 [90] South Africa | This intervention targeted mother–infant dyads from low socioeconomic environments in peri-urban areas to provide 16 60-min home visits. Sessions aimed to enhance parenting practices and attachment and were delivered by lay community workers, who implemented counselling strategies to provide mothers with psychological support and coaching on strategies to manage child distress. | Yes | Antenatal | 9 mo | Two-arm RCT (Mother) | Individual | Home | INT: 127 CTRL: 136 | X | | | | | X | | | X | X |
| Cronan et al., 1996 [91] USA | A program targeting low-income families participating in Head Start with children aged 1–3 years. Designed to promote early literacy, language, and communication skills through instructive shared book reading. The high-intensity intervention delivered 18 visits to parents, while the low intensity intervention provided 3 visits. | No | 27.8 mo | 9 mo | Three-arm RCT (Family) | Individual | Home | INT: 156 CTRL: 69 | X | X | | | | | | | | |
| Dozier et al., 2009 [92] USA | The Attachment and Biobehavioral Catch-up program is a home-based intervention that targets children and caregivers in foster homes. Ten weekly sessions supported caregivers to effectively interpret children's behaviors and provide nurturing care in order to promote attachment security. | Yes | 16.4 mo | 2.5 mo | Two-arm RCT (Child) | Individual | Home | INT: 22 CTRL: 24 | | | | | | X | | | | |
| Drotar et al., 2008 [93] USA | The Born to Learn program included home visits in the first month followed by monthly visits and group meetings that were structured to enhance child development. A trained parent educator delivered the visits, providing information on child development and supporting parents to enhance the quality of parent–child interactions. | Yes | 0 mo | 36 mo | Two-arm RCT (Family) | Individual, Group | Home, Community | INT: 141 CTRL: 154 | X | X | | X | | | | | | |
| Eddy et al., 2019 [94] USA | The Relief Nursery prevention program targets families with young children identified as at risk for child abuse and neglect. The program provides an integrated package of prevention services, including parent education and support, early childhood education, and home visiting, supplemented by other support services on an as-need basis. | No | 36 mo | 24 mo | Two-arm RCT (Family) | Individual, Group | Home, Early child education classroom, Community | INT: 90 CTRL: 90 | | | | | X | | | X | | X |
| Feil et al., 2019 [95] USA | Internet adaptation of the Play and Learning Strategies program, a preventive parenting intervention program to strengthen effective parenting practices through weekly video-based modeling of parenting skills and reflective coaching for parent–child dyads from low socioeconomic environments. | Yes | 4.4 mo | 6 mo | Two-arm RCT (Mother) | Individual | Virtual | INT: 80 CTRL: 73 | | X | | | | | X | | X | |

(Continued)

**Table 1.** (Continued)

| Author, Year Country | Intervention Description | Intervention Includes Responsive Caregiving | Average Child Age at Baseline | Average Program Duration | Study Design (Unit of Randomization) | Delivery | Setting | Analytic Sample | Early child development outcomes | | | | | | Parenting outcomes | | | |
|---|---|---|---|---|---|---|---|---|---|---|---|---|---|---|---|---|---|---|
| | | | | | | | | | Cognitive | Language | Motor | Socio-emotional | Behavior problems | Attachment | Knowledge | Practices | Parent-child interactions | Depression |
| Fergusson et al., 2005 [96] New Zealand | The Early Start Family Support intervention targeted families facing difficulties and stress. Family Support Workers conducted regular home visits to connect families with resources and work with the family to improve child health, reduce child abuse, improve parenting skills, support parental physical and mental health, and encourage positive partner relationships. | No | 2 mo | 36 mo | Two-arm RCT (Family) | Individual | Home | INT: 184 CTRL: 207 | X | | | | X | | | | | |
| Fernald et al., 2017 [97]; Knauer et al., 2016 [98] Mexico | The "Educación Inicial" (EI) program is a group-based parenting support program implemented among beneficiaries of the conditional cash transfer program Prospera. Weekly parenting group sessions were held on topics such as early childhood development, nutrition, responsive feeding to support the caregiver–child relationship. Parents in the first intervention group (T1) had access to both EI and Prospera, though there were no links between the programs. In the second group (T2), Prospera field staff received training on and encouraged parent participation in EI. | Yes | 9 mo | 48 mo | Three-arm Cluster RCT (Community) | Group | Community | INT: 747 CTRL: 366 | X | | | | | | | X | | |
| Field et al., 1982 [99] USA | This program targeted mothers in low-income environments. Mothers in the first intervention received biweekly home visits providing training in infant caretaking, the sensorimotor stage, and mother–infant interaction exercises. Mothers in the second intervention received on-the-job training in stimulation exercises as paid teacher-aides at a nursery program designed to improve infant's development, which their children also attended. | Yes | 0 mo | 6 mo | Three-arm RCT (Mother) | Individual, Group | Home, Childcare center | INT: 73 CTRL: 36 | X | | X | | | | | X | | |
| Frongillo et al., 2017 [100] Bangladesh | The Alive & Thrive intervention targeted families in rural Bangladesh. Nutrition-focused frontline workers provided regular home visits with counseling on responsive feeding and other infant and young child feeding (IYCF) messages and coached the mothers on relevant strategies and recommendations. Community social mobilization was organized, and a mass media campaign was implemented to broadcast IYCF messages through national and local media. | Yes | 22.3 mo | 41 mo | Two-arm Cluster RCT (Subdistrict) | Individual, Group | Home, Community, Media | INT: 1599 CTRL: 1603 | | X | X | | | | | | | |
| Galasso et al., 2019 [101] Madagascar | This study tested 4 intervention strategies combined with existing community-based services for growth monitoring and nutrition education. The intervention containing a parenting component comprised home visits by a community health worker to provide nutrition counseling, as well as fortnightly home visits for children aged 6–30 months to promote stimulation using a structured curriculum adapted from the Reach Up and Learn program. The other 3 intervention strategies included delivery of home visits for nutrition counselling alone, nutrition counseling plus provision of lipid-based nutrient supplementation (LNS) for children aged 6–18 months, and nutrition counseling plus provision of LNS for pregnant or lactating women. | No | Antenatal-12 mo | 24 mo | Five-arm Cluster RCT (Study site) | Individual | Home | INT: 732 CTRL: 736 | X | X | X | X | | | X | X | | |

*(Continued)*

**Table 1.** (Continued)

| Author, Year Country | Intervention Description | Intervention Includes Responsive Caregiving | Average Child Age at Baseline | Average Program Duration | Study Design (Unit of Randomization) | Delivery | Setting | Analytic Sample | Early child development outcomes | | | | | | Parenting outcomes | | | Depression |
|---|---|---|---|---|---|---|---|---|---|---|---|---|---|---|---|---|---|---|
| | | | | | | | | | Cognitive | Language | Motor | Socio-emotional | Behavior problems | Attachment | Knowledge | Practices | Parent-child interactions | |
| Gardner et al., 2005 [102] Jamaica | This intervention targeted children with weight-for-age Z scores below −1.5 with zinc deficiencies in order to supplement their development through psychosocial stimulation. The stimulation intervention consisted of weekly home visits conducted by trained community health workers to teach mothers how to engage with their child in an age-appropriate fashion. Simple toys were left with mothers each week. Mothers who received the nutrition intervention were given weekly zinc supplements. | Yes | 18.8 mo | 6 mo | 2 × 2 Factorial Cluster RCT (Clinic) | Individual | Home | INT: 46 CTRL: 68 | X | X | X | | | | | | | |
| Goldfeld et al., 2011 [103] Australia | Let's Read is a universal clinic-based literacy program targeting disadvantaged communities. Through 3 sessions delivered at age 4–8 months, 12 months, and 18 months, trained maternal and child health nurses modeled shared reading activities. | Yes | 2 mo | 14 mo | Two-arm Cluster RCT (Maternal and child health center) | Individual | Clinic | INT: 324 CTRL: 228 | | X | | | | | | X | | |
| Goodson et al., 2000 [104] USA | The Comprehensive Child Development Program is a two-generation program targeting families from low socioeconomic backgrounds. Home visits were delivered with 4 main activities: creation and follow-up of a family service plan based on a needs assessment, educating parents about child development and teaching parenting skills, developmental screening for all children, and referral and linkages with services. | No | 6 mo | 60 mo | Two-arm RCT (Family) | Individual, Group | Home, Community | INT: 1,256 CTRL: 1,301 | X | X | | | X | | | X | X | |
| Grantham-McGregor et al., 1991 [105] Jamaica | A psychosocial stimulation intervention was delivered to urban poor families with stunted children aged 9–24 months. The focus of the intervention was to teach mothers how to play with their children to promote development. Mothers receiving the nutrition intervention were given 1 kg milk-based formula per week. | No | 18.7 mo | 24 mo | 2 × 2 Factorial RCT (Child) | Individual | Home | INT: 62 CTRL: 65 | X | X | X | | | | | | | |
| Grantham-McGregor et al., 2020 [64] India | This study adapted the Reach-Up and Learn home visiting intervention. The facilitators used home visiting and group sessions to deliver messages about nutritional education and basic hygiene practices through games, stories, and cooking demonstrations. In the psychosocial stimulation component, facilitators showed mothers how to play and interact with and respond to their children in ways likely to promote development. Materials were handmade toys and purpose-designed books. Play materials were given to mothers to use at home and exchanged weekly. | Yes | 12 mo | 24 mo | Four-arm Cluster RCT (Village) | Individual, Group | Home, Community | INT: 655 CTRL: 643 | X | | X | X | X | | X | X | | |
| Guedeney et al., 2013 [106] France | The CAPEDP is a home-based intervention targeting vulnerable women in which a psychologist educated mothers on working alliance skills, early child development, attachment and health promotion and prevention during pregnancy. | No | Antenatal | 21 mo | Two-arm Cluster RCT (Hospital) | Individual | Home | INT: 184 CTRL: 183 | | | | | | X | | | | |
| Guttentag et al., 2014 [53] USA | The My Baby and Me intervention targeted high-risk mothers across 4 US states. The high-intensity home visitation coaching arm received 55 sessions with videotaped feedback sessions along with infant and toddler modules to teach mothers a set of interactional skills that collectively represent a responsive parenting style. The low-intensity arm included monthly phone calls from a coach, printed informational materials, and community resource referrals. | Yes | Antenatal | 33 mo | Two-arm RCT (Mother) | Individual | Home | INT: 97 CTRL: 128 | | X | | X | X | | | | X | |

*(Continued)*

**Table 1.** (Continued)

| Author, Year Country | Intervention Description | Intervention Includes Responsive Caregiving | Average Child Age at Baseline | Average Program Duration | Study Design (Unit of Randomization) | Delivery | Setting | Analytic Sample | Cognitive | Language | Motor | Socio-emotional | Behavior problems | Attachment | Knowledge | Practices | Parent-child interactions | Depression |
|---|---|---|---|---|---|---|---|---|---|---|---|---|---|---|---|---|---|---|
| | | | | | | | | | Early child development outcomes | | | | | | Parenting outcomes | | | |
| Hamadani et al., 2006 [107] Bangladesh | The Bangladesh Integrated Nutrition Program is a community-based intervention that provided nutrition supplementation to children with severe and moderate acute malnutrition. The psychosocial component of the intervention trained village women as play leaders to run group and individual meetings with mothers and children. Sessions promoted positive caring practices to improve mother–child interaction. Mothers were given low-cost picture books and information and demonstrations on the importance of play. | Yes | 14.8 mo | 12 mo | Two-arm Cluster RCT (Community nutrition center) | Individual, Group | Home, Community | INT: 92 CTRL: 101 | X | | X | | | | X | | | |
| Hamadani et al., 2019 [108] Bangladesh | This intervention targeted underweight children and aimed to enhance mothers' engagement in play and interactions with their child through a 25-session intervention delivered by government health workers in a primary health clinic setting. | Yes | 16 mo | 12 mo | Two-arm Cluster RCT (Clinic) | Group | Clinic | INT: 343 CTRL: 344 | X | X | X | X | | | X | X | | X |
| Hartinger et al., 2017 [109] Peru | This intervention adapted the Wawa Wasi National Program to be delivered to families in rural communities in a home-based setting. Mothers were trained in using the Wawa Wasi toys and materials to stimulate and interact with their children. Families were also provided age-appropriate toys. | Yes | 24 mo | 12 mo | Two-arm Cluster RCT (Community) | Individual | Home | INT: 219 CTRL: 216 | X | X | X | X | | | | | | |
| Heinicke et al., 1999 [110] USA | The UCLA Family Development Project targeted families identified as at risk for inadequate parenting during pregnancy. Mental health professionals delivered weekly home visits to mothers in the child's first year and biweekly in the second year. Mothers with infants aged 3–15 months had access to weekly mother–infant groups. Goals of the sessions were to reduce maternal depression, increase family support for mothers, and improve mother's responsiveness and infant security attachment. | Yes | Antenatal | 12 mo | Two-arm RCT (Family) | Individual, Group | Home | INT: 31 CTRL: 33 | X | | X | | | X | | X | X | X |
| Helmizar et al., 2017 [111] Indonesia | This community-based intervention consisted of a nutritional supplement intervention to deliver formula created from local food sources to families. Caregivers were given handbooks with instructions for preparing formula and information on complementary feeding. The psychosocial stimulation intervention received weekly parenting classes, guided by a handbook with 24 age-appropriate play sessions to enable mothers to play with their infants and support responsive mother–child interactions. | Yes | 7.7 mo | 6 mo | 2 × 2 Factorial Cluster RCT (Sub-village) | Group | Community | INT: 128 CTRL: 143 | X | X | X | | | | | X | | |
| Hepworth et al., 2020 [112] USA | The Attachment and Biobehavioral Catch-up intervention is an attachment-based intervention, implemented in the context of home-based Early Head Start visits, which provides coaching to increase parent sensitivity toward child early emotional regulation and to highlight efficacious strategies to prevent infant emotion dysregulation. Intervention activities include discussion of basic attachment principles, guided practice of new parenting behaviors, and review of video clips from previous sessions to reinforce parenting targets. | Yes | 12.7 mo | 3.3 mo | Two-arm RCT (Individual) | Individual | Home | INT: 76 CTRL: 72 | | | | | X | | | | X | |
| Heubner et al., 2000 [113] USA | A parent–child dialogic reading program consisting of 2 one-hour trainings in reading techniques delivered by children's librarians. Parents were trained to engage in reading behaviors through asking questions and expanding on child's utterances that in turn encouraged the child's verbal participation. | Yes | 28.7 mo | 1.5 mo | Two-arm RCT (Family) | Group | Community | INT: 78 CTRL: 36 | | X | | | | | | | X | |

*(Continued)*

**Table 1.** (Continued)

| Author, Year Country | Intervention Description | Intervention Includes Responsive Caregiving | Average Child Age at Baseline | Average Program Duration | Study Design (Unit of Randomization) | Delivery | Setting | Analytic Sample | Early child development outcomes | | | | | | Parenting outcomes | | | |
|---|---|---|---|---|---|---|---|---|---|---|---|---|---|---|---|---|---|---|
| | | | | | | | | | Cognitive | Language | Motor | Socio-emotional | Behavior problems | Attachment | Knowledge | Practices | Parent-child interactions | Depression |
| High et al., 2020 [114] USA | A literacy promoting intervention delivered by trained pediatric providers during well-child care visits, in which families received developmentally appropriate children's books and educational materials and advice about sharing books with children. Books contained brightly colored pictures and simple language, depicted culturally diverse images, and promoted parent–child interaction, while educational handouts presented benefits of reading to children beginning from a young age and focused on interaction between the parent and the child. | Yes | 7 mo | 11.5 mo | Two-arm RCT (Family) | Individual | Clinic | INT: 75 CTRL: 75 | | X | | | | | | X | | |
| Hutchings et al., 2017 [115] UK | The Incredible Years Toddler Parenting Program is a group-based parenting program targeting families living in highly disadvantaged areas. Twelve sessions were delivered by trained facilitators over a 1-year period. Group content focused on the importance of early childhood development, and parents who attended received free childcare services at community centers. | No | 21.3 mo | 6 mo | Two-arm RCT (Parent–child dyad) | Group | Community | INT: 60 CTRL: 29 | X | | | | X | | | X | X | X |
| Jacobs et al., 2016 [116] USA | The Healthy Families Massachusetts program targeted first-time adolescent parents. Through regular visits, program staff facilitated family goal setting and tailored support based on individual family needs. The program included developmental screenings, routine health screenings, and referrals to other services. | No | 0 mo | 14.7 mo | Two-arm RCT (Mother) | Individual | Home | INT: 358 CTRL: 236 | | | | | X | | | | | |
| Jin et al., 2007 [117] China | Targeted families in rural China and involved 2 counselling sessions using the Mother's Card model (adapted for China from WHO's Care for Development counselling materials). The sessions advised parents on how to stimulate their child's development and growth and provided advice for parents on age-appropriate communication with their child. | No | 10 mo | 6 mo | Two-arm RCT (Family) | Individual | Home | INT: 45 CTRL: 42 | | X | X | X | | | | | | |
| Johnson et al., 1974 [38] USA | This intervention targeted economically disadvantaged families. Mothers were enrolled in a 2-year parent education program designed to strengthen families and enable parents to optimize their child's mental, social and physical development. In year 1, a bilingual teacher provided weekly home visits for the mother and child and weekend sessions for the whole family. In year 2, the mother and child participated in a center-based nursery program, with evening sessions for fathers. The curriculum promoted mothers' home management skills and understanding of child development and sensitive responses. | Yes | 12 mo | 24 mo | Two-arm RCT (Family) | Individual, Group | Home | INT: 16 CTRL: 16 | X | | | | X | | | X | X | |
| Kaaresen et al., 2008 [118] Norway | This modified version of the Mother–Infant Transaction Program targeted low birth weight infants. Parents receive grief management sessions from a neonatal nurse, followed by 7 60-min sessions before discharge. Sessions aim to sensitize parents to the infant's cues and to coach parents how to respond appropriately. Four home visits were conducted by the same nurse after discharge to promote parents to interactively play with their child. | Yes | 0 mo | 3.25 mo | Two-arm RCT (Infant) | Individual | Hospital, Home | INT: 69 CTRL: 67 | X | | X | | X | | | | | |
| Kalinauskiene et al., 2009 [37] Lithuania | This home-based intervention (VIPP: video-feedback intervention to promote positive parenting) targeted mothers who rated low on sensitive responsiveness. During home-based sessions, mother–child interactions were videotaped and used to give feedback on how to sensitively respond to their child. | Yes | 6.1 mo | 5 mo | Two-arm RCT (Mother–infant dyad) | Individual | Home | INT: 26 CTRL: 28 | | | | | | X | | | X | X |

*(Continued)*

**Table 1.** (Continued)

| Author, Year Country | Intervention Description | Intervention Includes Responsive Caregiving | Average Child Age at Baseline | Average Program Duration | Study Design (Unit of Randomization) | Delivery | Setting | Analytic Sample | Cognitive | Language | Motor | Socio-emotional | Behavior problems | Attachment | Knowledge | Practices | Parent-child interactions | Depression |
|---|---|---|---|---|---|---|---|---|---|---|---|---|---|---|---|---|---|---|
| | | | | | | | | | **Early child development outcomes** | | | | | | | **Parenting outcomes** | | |
| Kaminski et al., 2013 [119] USA | The *Legacy* program targeted families living below the poverty line and in receipt of welfare assistance, beginning in pregnancy (Los Angeles) and at birth (Miami). The focus was to support parents to provide nurturing and supportive environments through strengthening self-efficacy for parenting behaviors, parenting support, and commitment to parenting. Topics included sensitive and responsive affection, discipline, routines, play, language promotion, and school readiness. | Yes | LA: Antenatal; Miami: 0 mo | LA: 36 mo; Miami: 60 mo | Two-arm RCT (Individual) | Individual, Group | Community, University campus | LA: INT: 126 CTRL: 78 Miami: INT: 121 CTRL: 73 | | | | X | X | | | | | |
| Khan et al., 2018 [120] Pakistan | An intervention targeting infants from poor urban localities, which aimed to improve mothers' ability to promote age-appropriate activities for ECD, improve child nutrition, and maternal mental health. Clinic assistants provided counseling for mothers during quarterly visits using a flip-book tool. | No | 0.5 mo | 6 mo | Two-arm Cluster RCT (Clinic) | Individual | Clinic | INT: 1,037 CTRL: 920 | X | X | X | X | | | | | | X |
| Kitzman et al., 1997 [121] USA | The Nurse Family Partnership program aims to improve outcomes for mothers and their children through home visiting. One group of mothers received intensive home visits from nurses during the antenatal period and 2 visits postpartum, along with screening and transportation services. A second group received the same services, plus continued home visits through child's age 24 months. Nurses provided information to support the mothers' own health, facilitate maternal–child interactions and help mothers understand how to communicate with their child to promote healthy development. | Yes | Antenatal | 30 mo | Four-arm RCT (Mother) | Individual | Home | INT: 206 CTRL: 465 | X | | | | X | | | X | X | X |
| Kochanska et al., 2013 [122] USA | The Child-Oriented Play program delivered training and play sessions to mothers from low-income backgrounds to provide coaching on how to respond to their child during play, follow their child's lead, focus attention on the child and allow the child to direct play. | Yes | 30.3 mo | 2.5 mo | Two-arm RCT (Mother–child dyad) | Invidual | Home, Lab | INT: 88 CTRL: 80 | | | | X | | | | | | |
| Kristensen et al., 2020 [123] Denmark | The Newborn Behavioral Observations system is an intervention to enhance caregivers' capacity to understand and respond sensitively to their newborn's cues. Health visitors conducted home visits to engage caregivers in observations of their infant and in shared dialogue to identify newborn behaviors and interpret these in the context of caregiver–infant interaction. | Yes | 0 mo | 3 mo | Two-arm Cluster RCT (Community) | Individual | Home | INT: 929 CTRL: 771 | | | | X | | | X | | X | X |
| Kyno et al., 2012 [124] Norway | The Mother-Infant Transaction Program targets preterm infants and consists of 11 60-min sessions, 7 predischarge, and 4 follow-up home visits. Sessions gave parents information about child development, their infant's uniqueness and developmental potential, child temperament, promoting social interactions, establishing routines and enjoyment of play. | Yes | 0 mo | 3.3 mo | Two-arm RCT (Child) | Individual | Hospital, Home | INT: 30 CTRL: 27 | X | X | X | X | | | | | | |
| Leung et al., 2017a [125] Hong Kong | The Fun to Learn for the Young program targeted children from disadvantaged backgrounds. Through 60 2-hour sessions, social workers and early childhood workers delivered content to parents consisting of: strategies to promote language, motor, social, and cognitive skills, behavior management strategies, and parenting skills. | Yes | 18 mo | 10 mo | Two-arm RCT (Parent–child dyad) | Group | Childcare center, Community | INT: 98 CTRL: 88 | X | | | X | X | | | | | |

(Continued)

**Table 1.** (Continued)

| Author, Year Country | Intervention Description | Intervention Includes Responsive Caregiving | Average Child Age at Baseline | Average Program Duration | Study Design (Unit of Randomization) | Delivery | Setting | Analytic Sample | Cognitive | Language | Motor | Socio-emotional | Behavior problems | Attachment | Knowledge | Practices | Parent-child interactions | Depression |
|---|---|---|---|---|---|---|---|---|---|---|---|---|---|---|---|---|---|---|
| | | | | | | | | | **Early child development outcomes** | | | | | | **Parenting outcomes** | | | |
| Leung et al., 2017b [126] Hong Kong | The Parent and Child Enhancement program targeted socioeconomically disadvantaged families. It aimed to enhance child learning and psychosocial development and equip parents with the skills and knowledge to promote the cognitive and psychosocial development of their child. It consists of 40 2-hour sessions, including information sharing with parents and direct teaching of children by social workers. | No | 27.6 mo | 5 mo | Two-arm RCT (Parent–child dyad) | Group | Community | INT: 76 CTRL: 73 | X | X | | | X | | | | | |
| Love et al., 2005 [127] USA | Early Head Start is a two-generation US federal program targeting low-income pregnant women and families with children up to the age of 3 years. This program provides comprehensive child development services through home visits, childcare, case management, parenting education or healthcare and referrals. | No | 5 mo | 24 mo | Two-arm RCT (Family) | Individual, Group | Home, Community | INT: 738 CTRL: 665 | X | X | | X | X | | | X | X | |
| Lozoff et al., 2010 [128] Chile | The intervention targeted 1 group of iron-deficient anemic and another group of nonanemic infants at either 6 or 12 months. The focus of the intervention was to promote mother–child relationships through learning about verbal and nonverbal shared activities, providing positive feedback, understanding child development, and supporting caregiver–child interactions. Data were extracted among the group of nonanemic infants. | Yes | 10.1 mo | 12 mo | Two-arm RCT (Infant) | Individual | Home | INT: 101 CTRL: 100 | X | | X | X | | | | | | |
| Luo et al., 2019 [129] China | An integrated home visitation program combining parental training in child psychosocial stimulation and child health promotion. Local community health workers provided training and education for caregivers on interactive caregiver–child activities to support child development and child health promotion, including appropriate child nutrition, hygiene habits, and other health-promoting behavior (e.g., oral hygiene of infants) by the primary caregiver. | No | 13 mo | 12 mo | Two-arm Cluster RCT (Village) | Individual | Home | INT: 190 CTRL: 200 | X | X | X | X | | | X | X | | |
| Madden et al., 1984 [130] USA | The Mother Child Hope Program targeted low-income families. This was a home-based early education intervention that delivered biweekly home visits, during which a Toy Demonstrator modeled and coached mothers in verbal interactions with the child, using toys and books that were given to the family. | Yes | 25 mo | 24 mo | Two-arm RCT (Family) | Individual | Home | INT: 25 CTRL: 23 | X | | | | | | | | X | |
| McGillion et al., 2017 [131] UK | This home-based intervention consisted of facilitators visiting families to show caregivers a video with information about the ways that infants indicate what they are interested in, along with examples of engaging in contingent talk with their child. Caregivers were asked to set aside 15 min/day to practice what they learned and monitor their child's behavior. | Yes | 11 mo | 1 mo | Two-arm RCT (Family) | Individual | Home | INT: 66 CTRL: 53 | | X | | | | | | | | |
| Mendelsohn et al., 2007 [132] USA | The Video Interaction Project (VIP) is a pediatric primary care program that targeted Latinx mothers and newborns. This program was delivered by a bilingual facilitator who coached mothers through feedback on individualized video sessions of mothers interacting with their newborns in order to promote early child development. | Yes | 0 mo | 33 mo | Two-arm RCT (Mother–infant dyad) | Individual | Clinic | INT: 52 CTRL: 45 | X | X | | | X | | | X | | X |

*(Continued)*

**Table 1.** (Continued)

| Author, Year Country | Intervention Description | Intervention Includes Responsive Caregiving | Average Child Age at Baseline | Average Program Duration | Study Design (Unit of Randomization) | Delivery | Setting | Analytic Sample | Early child development outcomes | | | | | | Parenting outcomes | | | |
|---|---|---|---|---|---|---|---|---|---|---|---|---|---|---|---|---|---|---|
| | | | | | | | | | Cognitive | Language | Motor | Socio-emotional | Behavior problems | Attachment | Knowledge | Practices | Parent-child interactions | Depression |
| Mendelsohn et al., 2018 [133] USA | The VIP is a pediatric primary care intervention promoting positive parenting through reading aloud and play. Bilingual facilitators use video feedback to reinforce responsive interactions and promote self-reflection. The intervention is delivered in 2 phases, (1) infant through toddler (VIP 0–3), and (2) preschool age. | Yes | 0.5 mo | 36 mo | Factorial RCT (Mother–infant dyad) | Individual | Clinic | INT: 153 CTRL: 143 | | | | X | X | | | | | |
| Muhoozi et al., 2017 [134]; Atukunda et al., 2019 [135] Uganda | This intervention was delivered by an education team of 4 trained individuals along with a village health team leader from the community. Three main sessions each lasting 6 to 8 hours were delivered to impoverished mothers, covering cooking demonstrations, nutrition education, hygiene and sanitation, and importance of play in child development. The village health team leader conducted additional follow-up sessions with mothers. | No | 7.4 mo | 6 mo | Two-arm Cluster RCT (Subcounty) | Group | Community | INT: 243 CTRL: 212 | X | X | X | X | | | | | | X |
| Nahar et al., 2012 [136]; Nahar et al., 2012b [137]; Nahar et al., 2015 [138] Bangladesh | This intervention targeted severely malnourished children from poor urban communities. In the psychostimulation intervention, play leaders provided center-based play sessions for mothers and children and parental education on child development. In the nutrition intervention, food packets were distributed, and mothers were instructed on how to prepare food. All participants received growth monitoring, health education, and micronutrient supplementation in the first 3 months. | Yes | 12.7 mo | 6 mo | Five-arm RCT (Child) | Individual, Group | Clinic | INT: 127 CTRL: 195 | X | | X | | | | | X | | X |
| Nair et al., 2009 [139] India | The Child Development Center model for early stimulation and therapy targets caregivers and infants during their first year of life. Mothers are trained on visual, auditory, tactile, and vestibular-kinesthetic stimulation by an occupational therapist at the center and encouraged to continue the exercises at home. | No | 0 mo | 12 mo | Two-arm RCT (Infant) | Individual, Group | Hospital, Home | INT: 324 CTRL: 341 | X | | X | | | | | | | |
| Norr et al., 2003 [140] USA | The REACH-Futures intervention targeted African- or Mexican-American pregnant women from low-income, inner city communities. A team of nurse-led community resident advocates conducted monthly home visits to discuss developmental concerns, parenting strategies, infant health nutrition, preventative care, and home safety. Nurses conducted developmental screening and health assessments during 3 home visits. | No | Antenatal | 12 mo | Two-arm RCT (Mother) | Individual | Home | INT: 258 CTRL: 219 | X | | X | | | | | X | | |
| Olds et al., 1986 [141] USA | The Nurse-Family Partnership program is delivered to vulnerable families beginning during pregnancy. The program provides parental education and support and linkages with local services. Topics include infant temperament, socioemotional development, learning and motor skills, responsive care, and physical and health needs of young children. | Yes | Antenatal | 27 mo | Four-arm RCT (Family) | Individual | Home | INT: 135 CTRL: 122 | X | | | | | | | X | | |
| Olds et al., 2002 [142] USA | A nurse visitation intervention (Nurse Family Partnership) in which nurses and paraprofessionals conducted home visits with mothers during pregnancy through their child's second birthday. Sessions included topics such as monitoring mother's diet and health, maternal substance use, developmentally appropriate play, creating safe household spaces, building supportive relationships, and establishing linkages to other services. | Yes | Antenatal | 28.5 mo | Three-arm RCT (Mother) | Individual | Home | INT: 480 CTRL: 255 | X | X | | | X | | | | X | |

*(Continued)*

 Parenting interventions for early child development: A systematic review and meta-analysis

**Table 1.** (Continued)

| Author, Year Country | Intervention Description | Intervention Includes Responsive Caregiving | Average Child Age at Baseline | Average Program Duration | Study Design (Unit of Randomization) | Delivery | Setting | Analytic Sample | Early child development outcomes | | | | | | Parenting outcomes | | | |
|---|---|---|---|---|---|---|---|---|---|---|---|---|---|---|---|---|---|---|
| | | | | | | | | | Cognitive | Language | Motor | Socio-emotional | Behavior problems | Attachment | Knowledge | Practices | Parent-child interactions | Depression |
| Pontoppidan et al., 2016 [143] Denmark | The Incredible Years Parents and Babies program trains group leaders to support parent groups by explaining the importance of brain development, demonstrating how parents can support their children's healthy development, and providing opportunities to practice new skills. | No | 1.6 mo | 2 mo | Two-arm RCT (Family) | Group | Community | INT: 67 CTRL: 35 | | | | X | | | | | X | X |
| Powell and Grantham-McGregor, 1989 [144] Jamaica | This intervention targeted children in low socioeconomic urban areas. Community health aides visited children weekly for hour-long visits. Each visit consisted of a combination of psychostimulation activities, such as language exercises, games, songs and crayon and paper activities. Information on child health and nutrition was provided to mothers. | Yes | 24.3 mo | 12 mo | Two-arm RCT (Child) | Individual | Home | INT: 29 CTRL: 29 | X | | X | | | | | | | |
| Powell et al., 2004 [145]; Baker-Henningham et al., 2005 [146] Jamaica | This intervention targeted undernourished children to deliver 30-min long weekly home visits, during which community health workers demonstrated and facilitated play activities with the mother and child. Community health workers discussed the importance of praise, attention and responsiveness, discipline strategies, child nutrition, and strategies to promote children's play and learning. | Yes | 18.4 mo | 12 mo | Two-arm Cluster RCT (Nutrition clinic) | Individual | Home | INT: 65 CTRL: 64 | X | X | X | | | | X | X | | X |
| Raby et al., 2019 [147] USA | Attachment and Biobehavioral Catch-up for Toddlers is a home visiting intervention that targets children in foster care. Parent coaches provide feedback to promote foster parents' sensitive and responsive behaviors in order to improve children's receptive vocabulary abilities. | Yes | 28.5 mo | 12 mo | Two-arm RCT (Parent) | Individual | Home | INT: 56 CTRL: 52 | | X | | | | | | | X | |
| Ramírez et al., 2019 [148] USA | This intervention provided parents of infants with coaching to improve parents' language inputs and social interactions with their child in order to enhance children's language development. One intervention group additionally attended group support sessions to share and discuss experiences about language interactions. | Yes | 6 mo | 4 mo | Three-arm RCT (Family) | Individual, Group | Lab, Community | INT: 53 CTRL: 24 | X | X | | | | | | | X | |
| Rauh et al., 1988 [149] USA | The Mother-Infant Transaction Program targeted low birth weight infants. Through 7 hospital-based and 4 home-based sessions, a neonatal intensive care nurse supported mothers to read and appropriately respond to their infant's individual cues. Mothers were also coached on how to provide interactive play experiences for their child. | Yes | 0 mo | 3.3 mo | Two-arm RCT (Mother–infant dyad) | Individual, Group | Home, Hospital | INT: 25 CTRL: 28 | X | | | | | | | | | |
| Robling et al., 2016 [150] UK | Family Nurse Partnership program targeted adolescent mothers. Through a maximum of 64 one-hour home visits, trained family nurses provided mothers with content on antenatal health behaviors, parenting, and caregiving of young infants and toddlers. | Yes | Antenatal | 27 mo | Two-arm RCT (Mother) | Individual | Home | INT: 480 CTRL: 415 | X | X | | | | | | | X | |
| Rockers et al., 2016 [151] Zambia | A community-based intervention in which Child Development Agents conduct fortnightly home visits to screen young children for health and nutrition needs. Child Development Agents made recommendations on seeking routine child services, facilitated group meetings to discuss parenting skills, child nutrition, cooking, and activities to promote child development. | No | 8.5 mo | 12 mo | Two-arm Cluster RCT (Health zone) | Individual, Group | Home, Community | INT: 220 CTRL: 215 | X | | X | | | | | X | | X |
| Roggman et al., 2009 [152] USA | The Bear River Early Head Start program provided weekly home visits and peer groups for pregnant women from low-income families and their children during the first 3 years of life. The weekly home visits aimed to help parents foster responsive parent–child interactions, enhance parents' understanding of their children's development and promote attachment security. | Yes | 10 mo | 26 mo | Two-arm RCT (Mother) | Individual, Group | Home, Community | INT: 77 CTRL: 78 | X | | | | | X | | | | |

(Continued)

**Table 1.** (Continued)

| Author, Year Country | Intervention Description | Intervention Includes Responsive Caregiving | Average Child Age at Baseline | Average Program Duration | Study Design (Unit of Randomization) | Delivery | Setting | Analytic Sample | Early child development outcomes Cognitive | Language | Motor | Socio-emotional | Behavior problems | Attachment | Parenting outcomes Knowledge | Practices | Parent-child interactions | Depression |
|---|---|---|---|---|---|---|---|---|---|---|---|---|---|---|---|---|---|---|
| Santelices et al., 2010 [153] Chile | The Promoting Secure Attachment Program targeted pregnant mothers from the second trimester until the first year of the baby's life. The first phase consisted of group workshops for pregnant mothers to discuss mother–child attachment. The second phase consisted of individual sessions on mother–child dyads to provide feedback on mothers' sensitive responsiveness. | Yes | Antenatal | 18 mo | Two-arm RCT (Mother) | Individual, Group | Home, Clinic | INT: 43 CTRL: 29 | | | | | | X | | | | |
| Sawyer et al., 2017 [154] Australia | An intervention targeting mother–child dyads with infants aged 1–7 months, which comprised clinic-based postnatal health check-ups and a nurse-moderated internet-based support group. | No | 1 mo | 6 mo | Two-arm RCT (Mother) | Individual, Group | Clinic, Virtual | INT: 240 CTRL: 251 | | | | X | | | | | | |
| Scarr and McCartney, 1988 [155] Bermuda | The Mother Child Hope Program is a weekly home-based intervention that provided families with toys or books and demonstrated to caregivers how to engage in play behaviors with their child. | No | 27 mo | 24 mo | Two-arm RCT (Mother–child dyad) | Individual | Home | INT: 78 CTRL: 39 | X | | | | | | | | | |
| Schaub et al., 2019 [156] Switzerland | The Parents as Teachers is a home visiting program delivered by parent educators which aims to increase of parental knowledge of early childhood development and improve parental practices, including early detection of developmental delays and health issues, prevention of child abuse and neglect, and a children's longer-term school readiness and success. Parents also may attend optional monthly group meetings to build parenting networks and receive supplementary information. | No | 1.7 mo | 36 mo | Two-arm RCT (Family) | Individual, Group | Home, Community | INT: 109 CTRL: 102 | X | X | X | | X | | | | | |
| Schwarz et al., 2012 [157] USA | The home-based intervention provided caregivers 11 home visits that aimed to promote their child's growth and development, as well as to provide resources to access primary healthcare and early intervention services. | No | 0 mo | 36 mo | Two-arm RCT (Mother) | Individual | Home | INT: 135 CTRL: 134 | X | X | | | | | | | | X |
| Shi et al., 2020 [158] China | This study integrated a parenting intervention within existing primary health care services. Caregivers in the intervention group received a parenting pamphlet and 2 parenting training sessions during well-child clinic visits. Caregivers with children with suspected developmental delays received additional parenting guidance by telephone. | No | 2 mo | 12 mo | Two-arm Cluster RCT (Community) | Individual | Clinic | INT: 71 CTRL: 69 | X | X | X | X | | | | | | |
| Singla et al., 2015 [36] Uganda | This intervention targeted mother–child dyads in rural communities to engage participants in group sessions delivering content including use of play materials, gentle discipline, consumption of a diverse diet, hygiene and sanitation messages, and maternal well-being. | Yes | 22.3 mo | 7 mo | Two-arm Cluster RCT (Parish) | Individual, Group | Home, Community | INT: 160 CTRL: 131 | X | X | | | | | X | X | | |
| Slade et al., 2019 [159] USA | Minding the Baby is an attachment-based, home visiting intervention targeting young first-time mothers living in underserved urban communities. It aims to support mothers to develop higher levels of parental reflective functioning, to minimize levels of disrupted affective communication, to promote secure attachment, and to have a positive impact on maternal depression and PTSD. | Yes | Antenatal | 27 mo | Two-arm RCT (Mother) | Individual | Home | INT: 60 CTRL: 64 | | | | | | X | | | X | X |

(Continued)

**Table 1.** (Continued)

| Author, Year Country | Intervention Description | Intervention Includes Responsive Caregiving | Average Child Age at Baseline | Average Program Duration | Study Design (Unit of Randomization) | Delivery | Setting | Analytic Sample | Cognitive | Language | Motor | Socio-emotional | Behavior problems | Attachment | Knowledge | Practices | Parent-child interactions | Depression |
|---|---|---|---|---|---|---|---|---|---|---|---|---|---|---|---|---|---|---|
| | | | | | | | | | Early child development outcomes | | | | | | Parenting outcomes | | | |
| Spieker et al., 2012 [160] USA | This intervention randomized participants to receive either the Promoting First Relationships (intervention) or Early Education Support (control) programs. Participants in Promoting First Relationships received 10 weekly sessions from a trained graduate student who provided parents with feedback using videotaped caregiver–child interactions, including discussions about interpretation of child cues. Early Education Support participants received 3 monthly 90-min visits from trained individuals who connected caregivers to resources, referrals, and activities to stimulate child growth and development. | Yes | 18 mo | 2.5 mo | Two-arm RCT (Child) | Individual | Home | INT: 86 CTRL: 89 | | | | X | X | X | X | | X | X |
| Tofail et al., 2013 [161] Bangladesh | This intervention was delivered to 2 groups of infants aged 6–24 months either with iron deficiency anemia or those who were not anemic or iron deficient in rural villages. Children were provided nutritional and psychosocial resources. The nutrition component included weekly liquid iron supplements for the first 6 months, and the psychosocial component included weekly home visits that focused on promoting mother–child interactions and play. Data were extracted among the group of nonanemic infants. | Yes | 16.1 mo | 9 mo | Two-arm Cluster RCT (Village) | Individual | Home | INT: 104 CTRL: 92 | X | | X | | | | | X | | |
| Vally et al., 2015 [162]; Murray et al., 2016 [55] South Africa | This intervention provided mothers in rural communities with the resources to learn how to practice book sharing with their infants and responsively interact and communicate with their children. | Yes | 15.4 mo | 2 mo | Two-arm RCT (Carer–infant dyad) | Group | Community | INT: 45 CTRL: 37 | | X | | | | | | X | X | |
| Van Zeijl et al., 2006 [35] the Netherlands | The Video-feedback Intervention to promote Positive Parenting and Sensitive Discipline was delivered to families with children who exhibited externalizing behavior. The intervention arm included videotaping and providing feedback on parent–child interactions and sharing supplemental information about child development. | Yes | 27 mo | 8 mo | Two-arm RCT (Family) | Individual | Home | INT: 120 CTRL: 117 | | | | | X | | | | X | |
| Vazir et al., 2013 [163] India | This intervention enrolled families in rural communities to target child diet, growth, and development. Families in the first intervention group received Integrated Child Development Services (ICDS) and home visits comprising of nutrition education, responsive feeding, and appropriate play-based stimulation messages. Families in a second intervention group received ICDS services and home visits consisting of nutrition education only. | Yes | 3 mo | 12 mo | Three-arm Cluster RCT (Village) | Individual | Home | INT: 329 CTRL: 182 | X | | X | | | | | | | X |
| Velderman et al., 2006 [164] the Netherlands | The VIPP intervention was delivered to first-time mothers and aimed to promote maternal sensitivity through sharing information about sensitive parenting and providing personalized video feedback about mothers' interactions with their child. A second intervention, VIPP-R, additionally included discussions of mothers' childhood attachment experiences in relation to their current caregiving. | Yes | 6.8 mo | 5 mo | Three-arm RCT (Mother) | Individual | Home | INT: 54 CTRL: 27 | | | | | | X | | | X | |
| Waber et al., 1981 [165] Colombia | The nutrition intervention group received daily food supplements, including bread, milk, protein, vitamins, and minerals. The maternal education intervention included home visits by trained individuals that provided mothers with information on child development and responsive caregiving. | Yes | Antenatal | 36 mo | Six-arm RCT (Mother) | Individual | Home | INT: 76 CTRL: 111 | X | | | | | | | | | |

(Continued)

**Table 1.** (Continued)

| Author, Year Country | Intervention Description | Intervention Includes Responsive Caregiving | Average Child Age at Baseline | Average Program Duration | Study Design (Unit of Randomization) | Delivery | Setting | Analytic Sample | Early child development outcomes | | | | | | Parenting outcomes | | | |
|---|---|---|---|---|---|---|---|---|---|---|---|---|---|---|---|---|---|---|
| | | | | | | | | | Cognitive | Language | Motor | Socio-emotional | Behavior problems | Attachment | Knowledge | Practices | Parent-child interactions | Depression |
| Wagner et al., 2002 [166] USA | The Parents as Teachers intervention targeted low-income families and encouraged positive child development through parent education. Education was provided through regular home visits and group meetings by trained parent educators with the aim of increasing parents' knowledge of child development, preventing child abuse, and supporting parental feelings of competence. | No | 4.5 mo | 24 mo | Two-arm RCT (Family) | Individual, Group | Home, Community | INT: 275 CTRL: 390 | X | X | X | X | | | X | X | X | |
| Wake et al., 2011 [167] Australia | Let's Learn Language is a modified version of the You Make the Difference program designed to target infants identified as at-risk of language delay. It promotes responsive interactions and language. | Yes | 13.3 mo | 1.5 mo | Two-arm Cluster RCT (Maternal and child health center) | Group | Community | INT: 135 CTRL: 133 | | X | | | X | | | | | |
| Walker et al., 2004 [168] Jamaica | A home visiting intervention that targeted preterm low birth weight infants in 2 phases. The first phase during the child's first 8 weeks of life consisted of weekly 60-min visits from community health workers to teach parents activities to promote early child development. Phase 2 occurred between 7–24 months of age, during which community health workers demonstrated play techniques to the mother and engaged her in play sessions with her child. | Yes | 0 mo | 24 mo | Two-arm RCT (Infant) | Individual | Home | INT: 63 CTRL: 67 | X | | | | | | | X | | |
| Walkup et al., 2009 [34] USA | A home visiting intervention entitled Family Spirit that delivered content to young American Indian mothers on prenatal and newborn care and maternal life skills. The goal of this intervention was to increase mothers' parenting knowledge, involvement, and stimulation of the child in home environments. | No | Antenatal | 9 mo | Two-arm RCT (Mother) | Individual | Home | INT: 37 CTRL: 45 | | | | X | X | | X | X | | X |
| Wallander et al., 2014 [169] India, Pakistan, Zambia | A home-based intervention following the Partners for Learning curriculum targeting disadvantaged infants exposed and not exposed to birth asphyxia. The intervention goal was to promote early development by training caregivers to stimulation their children using developmentally and age-appropriate play and learning activities. | No | 1 mo | 36 mo | Two-arm RCT (Infant) | Individual | Home | INT: 185 CTRL: 191 | X | | X | | | | | | | |
| Wasik et al., 1990 [170] USA | The Family Education Program is a home-based intervention for parents with disadvantaged educational or social circumstances. The intervention aims to increase effective parenting by providing developmentally appropriate activities and problem solving and goal setting skills. | No | 1 mo | 36 mo | Three-arm RCT (Family) | Individual, Group | Home, Childcare center | INT: 25 CTRL: 23 | X | | | | | | | X | | |
| Weisleder et al., 2016 [171] USA | The Bellevue Project for Early Language, Literacy and Education Success intervention aims to promote socioemotional development for the parent and child through either individualized sessions that utilized video recordings to provide feedback on parent–child interactions or by providing parents with a newsletter on developmentally appropriate interactions. | Yes | 0 mo | 36 mo | Three-arm RCT (Mother–infant dyad) | Individual | Clinic | INT: 152 CTRL: 149 | | | | X | X | | | | | |
| Whitt and Casey, 1982 [172] USA | The goal of this intervention was to enhance infants' cognitive development through promoting the mother's awareness of her infant's social nature, improving the mother's knowledge of infant development, and increasing maternal feelings of confidence and competence to affect her child's development. | Yes | 0.5 mo | 6 mo | Three-arm RCT (Mother–infant dyad) | Individual | Clinic | INT: 15 CTRL: 17 | X | | | | | | | | X | |

*(Continued)*

**Table 1.** (Continued)

| Author, Year Country | Intervention Description | Intervention Includes Responsive Caregiving | Average Child Age at Baseline | Average Program Duration | Study Design (Unit of Randomization) | Delivery | Setting | Analytic Sample | Early child development outcomes | | | | | | Parenting outcomes | | | Depression |
|---|---|---|---|---|---|---|---|---|---|---|---|---|---|---|---|---|---|---|
| | | | | | | | | | Cognitive | Language | Motor | Socio-emotional | Behavior problems | Attachment | Knowledge | Practices | Parent-child interactions | |
| Yousafzai et al., 2014 [173]; Yousafzai et al., 2015 [174] Pakistan | A responsive stimulation and nutrition intervention that aimed to promote child developmental outcomes. Using an adapted version of the Care for Child Development approach, Lady Health Workers promoted caregiver sensitivity, responsiveness, and developmentally appropriate play between caregiver and child. | Yes | 0.7 mo | 24 mo | 2 × 2 Factorial Cluster RCT (Lady Health Worker catchment area) | Individual, Group | Home, Community | INT: 657 CTRL: 641 | X | X | X | X | | | X | X | X | X |

CTRL, control group; ECD, early child development; EI, Educación Inicial; ICDS, Integrated Child Development Services; INT, intervention group; RCT, randomized controlled trial; RFS, Responsive Feeding and Stimulation; VIP, Video Interaction Project; VIPP, video-feedback intervention to promote positive parenting;

and India (5 trials). Studies were published between 1974 and 2020. Analytic sample sizes ranged from 32 to 3,202 individuals. The average age of children at baseline varied across interventions: 16 trials (16%) enrolled pregnant mothers, 30 (29%) enrolled children with an average age of 3 months and younger, 24 (24%) enrolled children with an average age of 3 to 12 months, and 32 (31%) enrolled children who were on average older than 12 months. Intervention duration ranged from a brief intervention delivered over 1 month to longer interventions delivered across the first 5 years of the child's life. Nearly all interventions (93%) focused on supporting mothers exclusively, with only 7 interventions (7%) additionally engaging fathers to some degree [34–40].

Out of the total 102 interventions, 70 (69%) had some degree of a responsive caregiving component that aimed to improve parent–child interactions. Of these responsive caregiving interventions, 37 interventions (53%) involved facilitators who observed live parent–child interactions and/or utilized video recordings in order to provide feedback for promoting parental sensitivity and responsiveness in the context of play, communication, reading, and/or feeding. The other 33 responsive caregiving interventions (47%) more generally aimed to improve the quality of parent–child interactions through guidance, demonstrations, and encouragement of parents to practice during sessions and on their own.

The remaining 32 interventions (31%) did not include responsive caregiving components. Of these, 22 (69%) provided general parenting support with information and guidance on child development and parenting topics, and 10 (31%) focused on increasing parental stimulation and/or provision of learning materials. Among the total 102 interventions, 45 (44%) also integrated messages beyond parenting for ECD, such as maternal and child nutrition, child health, caregiver mental health, and early education.

Interventions utilized a range of delivery models: 58 interventions (57%) were delivered to individual caregivers and families, 13 (13%) were exclusively group based, and 31 (30%) incorporated both individually delivered and group-based components. Delivery settings also varied. Nearly half or 47 interventions (46%) were delivered exclusively through home visits, 11 in a clinical or hospital context (11%), 11 in a community space (11%), and 32 interventions (31%) used a combination of multiple settings. One intervention was delivered virtually.

## Risk of bias assessment

The total risk of bias score across all studies was moderate (mean = 5.1, SD = 1.5) ranging from 2 to 9 on a scale of 0 to 16. In general, risk of bias was low for random sequence generation, blinding of outcome assessors, and incomplete outcome data (S1 Fig). However, risk of bias for allocation concealment and selective reporting were unclear for the majority of studies. Given the nature of psychoeducational and behavioral parenting interventions that involve parents' active participation, blinding of participants was not possible. Egger's tests suggested evidence of small-sample bias for 2 out of the 10 outcomes: child language development, $z = 2.74$, $P = 0.01$; and parent–child interactions, $z = 3.78$, $P < 0.001$.

## Effects on early child development outcomes

Fifty-eight studies provided a total of 58 effect sizes for children's cognitive development. Approximately half of these studies (56%) reported any details regarding reliability or validity of the measure of cognitive development. The Bayley Scales of Infant and Toddler Development (BSID) [41] was the most commonly used measure in more than half the studies (54%). See S2 Table for full list of measures used for each of the outcomes. The pooled result showed a moderate positive impact of parenting interventions on improving cognitive development (SMD = 0.32, 95% CI: 0.23, 0.40, $P < 0.001$; $I^2 = 89\%$, $P < 0.001$; Fig 2).

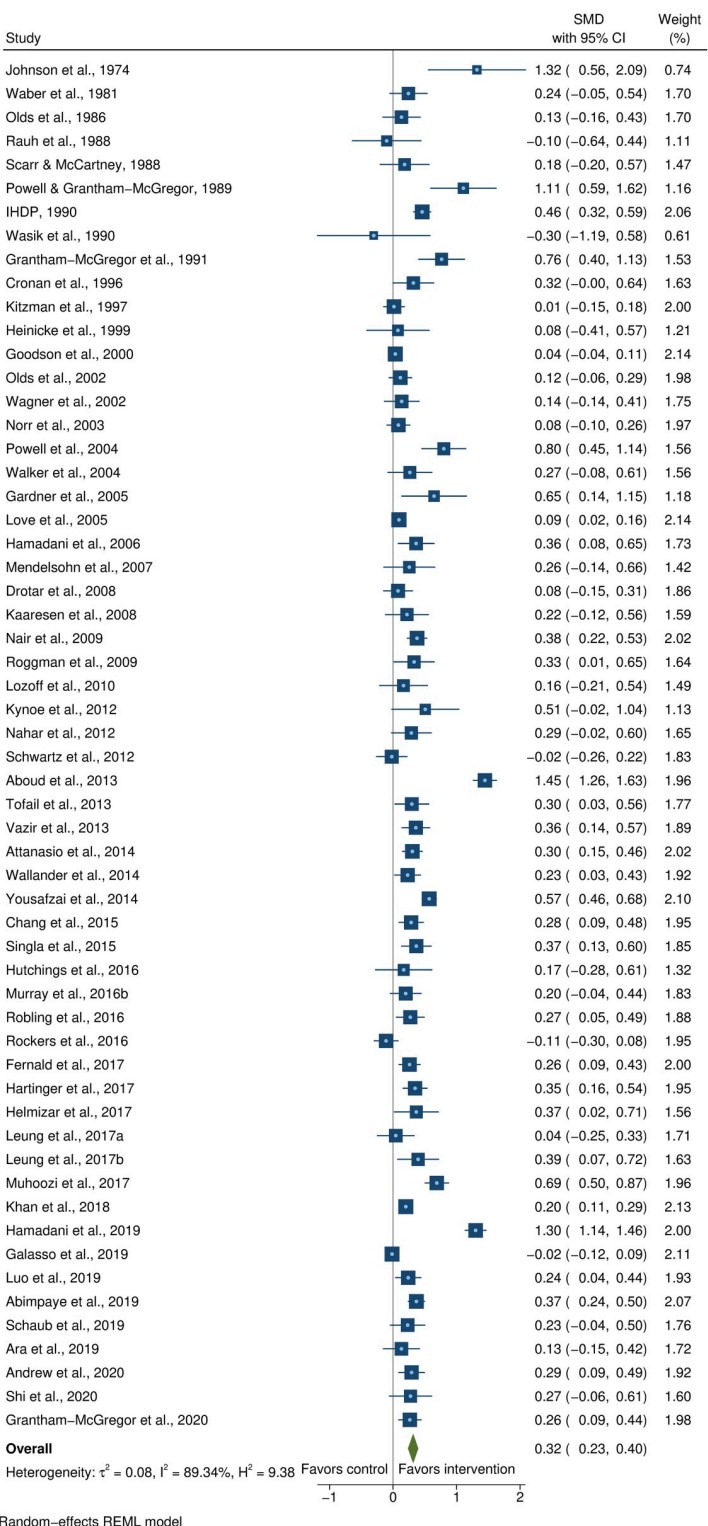

**Fig 2. Forest plot for the effect of parenting interventions on cognitive development.** Blue squares represent the SMD for each study, with the size of the square being proportional to the study weight. The whiskers extending from each side of the square represent the range of the 95% CI. The green diamond shows the overall pooled effect size using a random-effects model, which is centered at the point estimate and the diamond width representing the 95% CI. SMD, standardized mean difference.

Forty-six studies provided a total of 58 effect sizes for children's language development. Approximately half of these studies (57%) reported any details regarding reliability or validity of the measure of language development. The BSID (25%) and the MacArthur-Bates Communicative Development Inventory (14%) were the most commonly used measures. The pooled result showed a moderate positive impact on improving language development (SMD = 0.28, 95% CI: 0.18, 0.37, $P < 0.001$; $I^2$ = 91%, $P < 0.001$; Fig 3).

Thirty-five studies provided a total of 45 effect sizes for children's motor development. Approximately half of these studies (56%) reported any details regarding reliability or validity of the measure of motor development. The BSID was the most commonly used measure in more than half the studies (56%). The pooled result showed a moderate positive impact on improving motor development (SMD = 0.24, 95% CI: 0.15, 0.32, $P < 0.001$; $I^2$ = 84%, $P < 0.001$; Fig 4).

Twenty-nine studies provided a total of 31 effect sizes for children's socioemotional development. Nearly 2 in 5 studies (38%) reported any details regarding reliability or validity of the measure of socioemotional development. All measures were caregiver reported and most commonly included the Ages and Stages Questionnaire [42] (16%), Ages and Stages Questionnaire: Social-Emotional [43] (13%), and the Brief Infant–Toddler Social and Emotional Assessment [44] (13%). The pooled result showed a small positive impact on improving socioemotional development (SMD = 0.19, 95% CI: 0.10, 0.28, $P < 0.001$; $I^2$ = 81%, $P < 0.001$; Fig 5).

Twenty-nine studies provided a total of 35 effect sizes for children's behavior problems. Nearly half of the studies (47%) reported any details regarding reliability or validity of the measure of behavioral problems. All measures were caregiver reported, and the Child Behavior Checklist [45] was the most commonly used measure in more than half the studies (53%). The pooled result showed a small impact on reducing behavior problems (SMD = −0.13, 95% CI: −0.18, −0.08, $P < 0.001$; $I^2$ = 53%, $P < 0.001$; Fig 6).

Eleven studies provided a total of 11 effect sizes for infant–caregiver attachment. The majority of studies (85%) reported any details regarding reliability or validity of the measure of attachment. The Ainsworth Strange Situation Procedure [46] was the most commonly used measure in 2 out of 5 studies (38%). The pooled result showed a moderate positive impact on improving attachment (SMD = 0.29, 95% CI: 0.18, 0.40, $P < 0.001$; $I^2$ = 0%, $P = 0.92$; Fig 7).

### Effects on parenting outcomes

Sixty-four studies (63%) measured at least 1 parenting outcome, and 35 studies (34%) measured 2 or more parenting outcomes. Sixteen studies provided a total of 16 effect sizes for parenting knowledge. Less than half of studies (44%) reported any details regarding reliability or validity of the measure of parenting knowledge. Nearly all studies (88%) used a knowledge questionnaire that was developed by the authors. The pooled result showed a large positive impact on improving parenting knowledge (SMD = 0.56, 95% CI: 0.33, 0.79, $P < 0.001$; $I^2$ = 97%, $P < 0.001$; Fig 8).

Thirty-five studies provided a total of 40 effect sizes for parenting practices. Two-thirds of studies (66%) reported any details regarding reliability or validity of the measure of parenting practices. The Home Observation of Measurement of the Environment Inventory-Infant and Toddler version [47] was the most commonly used measure in more than half of studies (58%). The pooled result showed a moderate positive impact on improving parenting practices (SMD = 0.33, 95% CI: 0.22, 0.44, $P < 0.001$; $I^2$ = 93%, $P < 0.001$; Fig 9).

Twenty-seven studies provided a total of 27 effect sizes for parent–child interactions. Three in four studies (77%) reported any details regarding reliability or validity of the measure of parent–child interactions. Nearly all studies directly observed and coded quality of mother–child

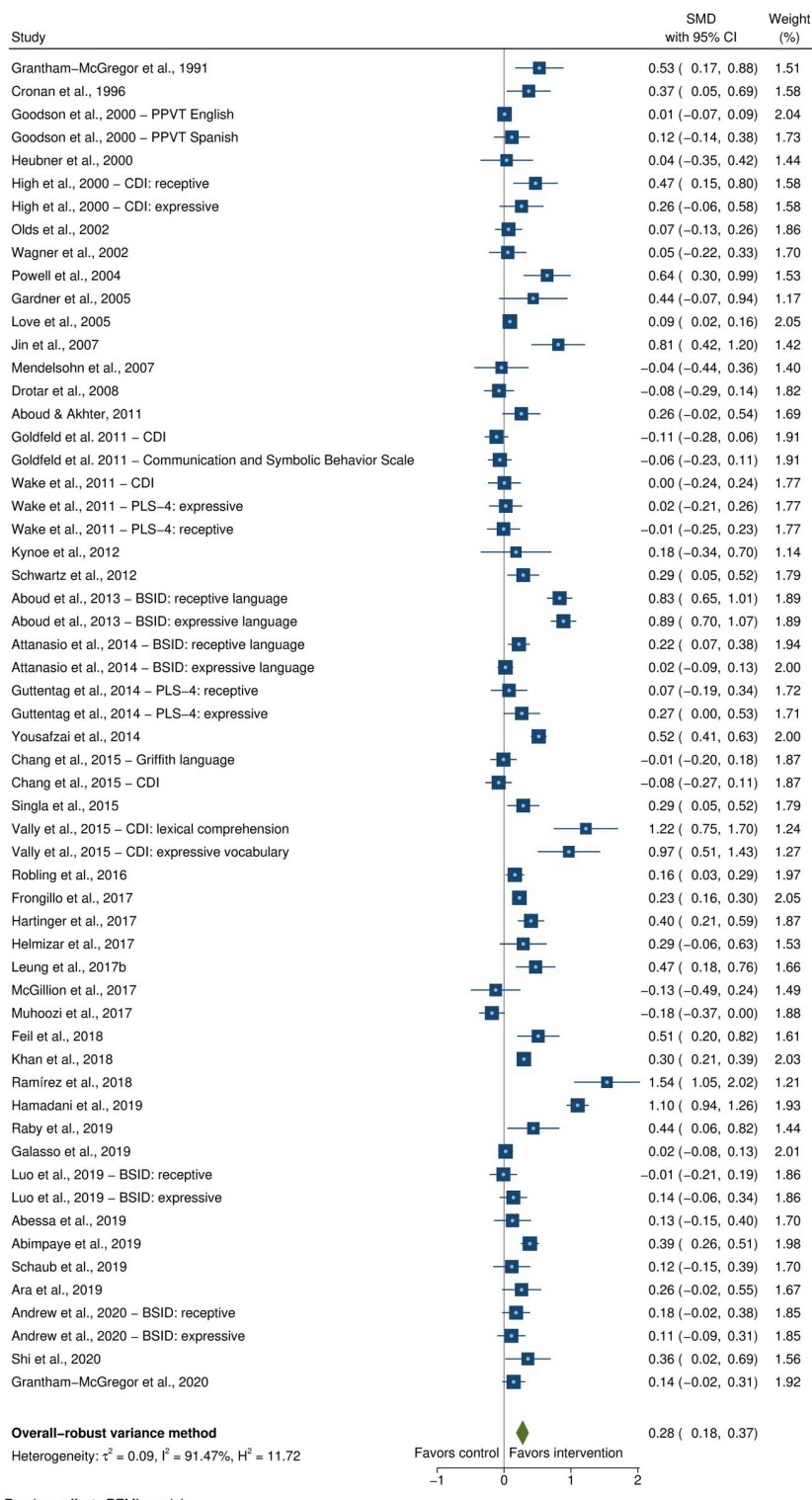

**Fig 3. Forest plot for the effect of parenting interventions on language development.** Blue squares represent the SMD for each study, with the size of the square being proportional to the study weight. The whiskers extending from each side of the square represent the range of the 95% CI. The green diamond shows the overall pooled effect size using a random-effects model, which is centered at the point estimate and the diamond width representing the 95% CI. SMD, standardized mean difference.

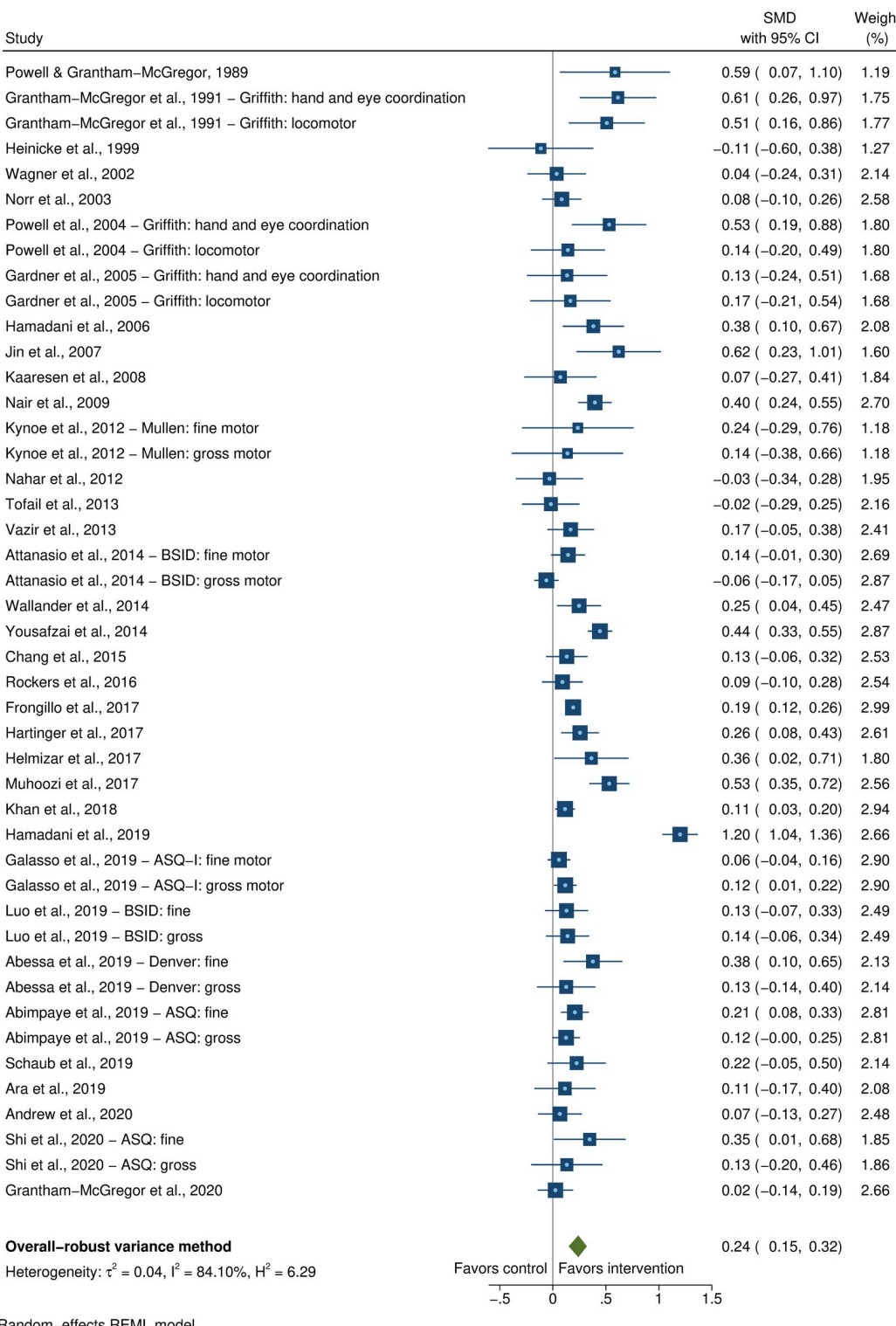

**Fig 4. Forest plot for the effect of parenting interventions on motor development.** Blue squares represent the SMD for each study, with the size of the square being proportional to the study weight. The whiskers extending from each side of the square represent the range of the 95% CI. The green diamond shows the overall pooled effect size using a random-effects model, which is centered at the point estimate and the diamond width representing the 95% CI. SMD, standardized mean difference.

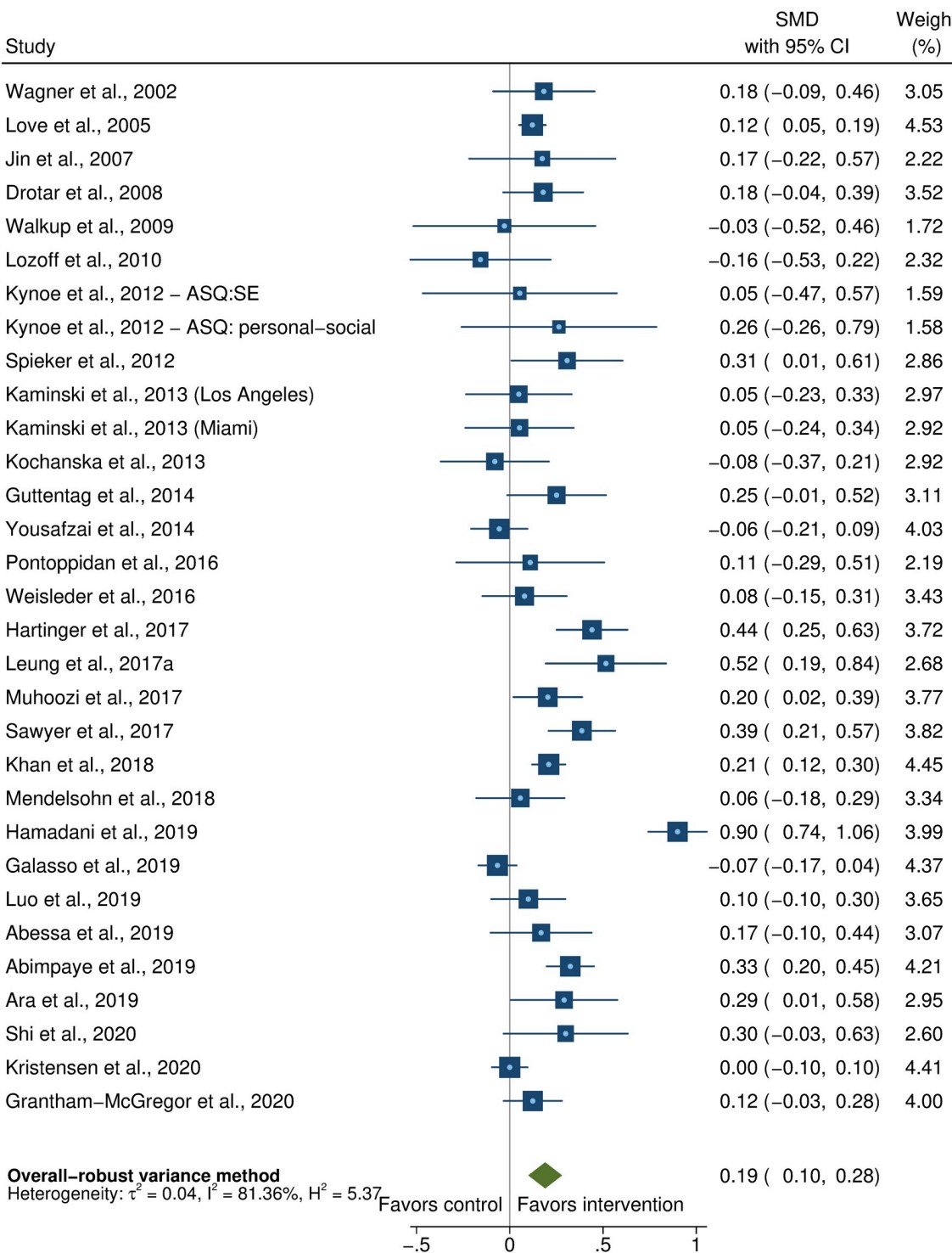

**Fig 5. Forest plot for the effect of parenting interventions on socioemotional development.** Blue squares represent the SMD for each study, with the size of the square being proportional to the study weight. The whiskers extending from each side of the square represent the range of the 95% CI. The green diamond shows the overall pooled effect size using a random-effects model, which is centered at the point estimate and the diamond width representing the 95% CI. SMD, standardized mean difference.

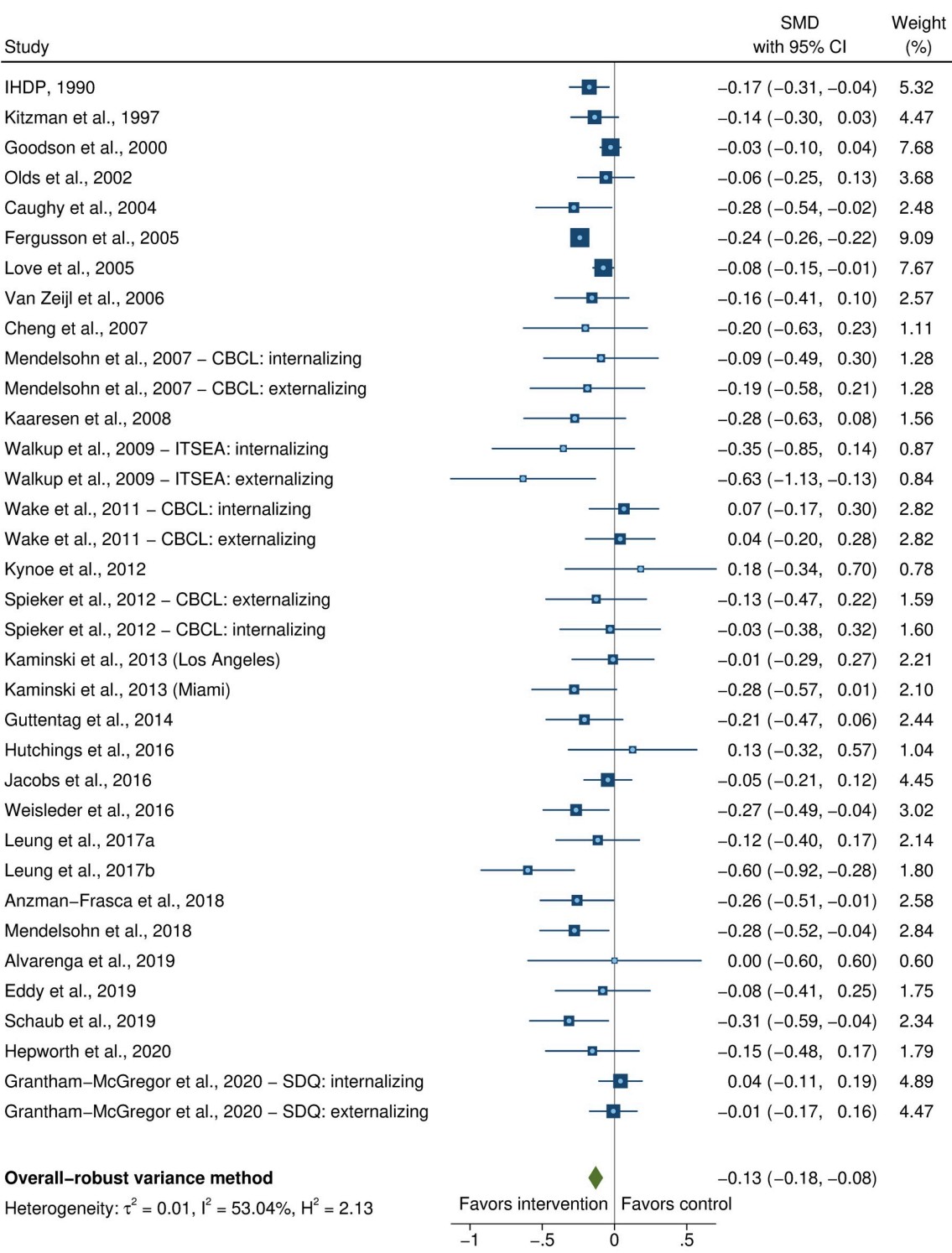

**Fig 6. Forest plot for the effect of parenting interventions on behavior problems.** Blue squares represent the SMD for each study, with the size of the square being proportional to the study weight. The whiskers extending from each side of the square represent the range of the 95% CI. The green diamond shows the overall pooled effect size using a random-effects model, which is centered at the point estimate and the diamond width representing the 95% CI. SMD, standardized mean difference.

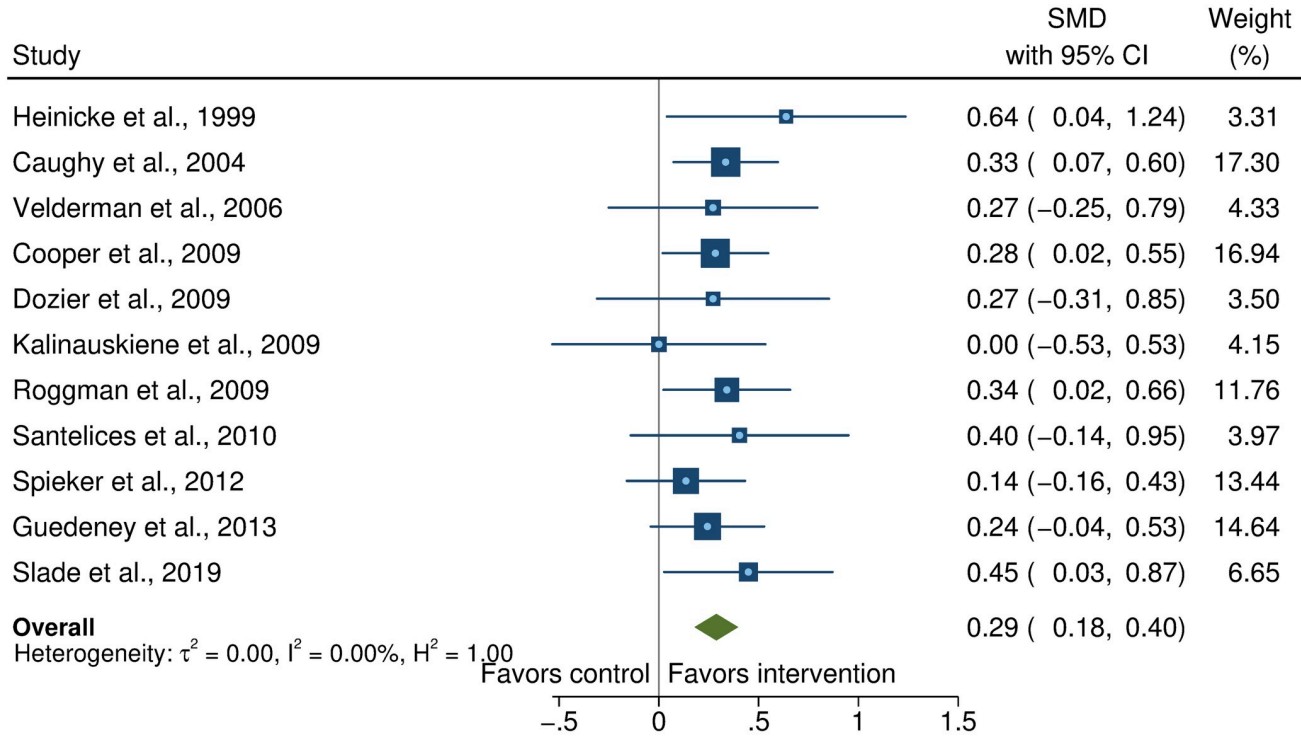

**Fig 7. Forest plot for the effect of parenting interventions on attachment.** Blue squares represent the SMD for each study, with the size of the square being proportional to the study weight. The whiskers extending from each side of the square represent the range of the 95% CI. The green diamond shows the overall pooled effect size using a random-effects model, which is centered at the point estimate and the diamond width representing the 95% CI. SMD, standardized mean difference.

interactions (91%). The pooled result showed a moderate positive impact on improving parent–child interactions (SMD = 0.39, 95% CI: 0.24, 0.53, $P < 0.001$; $I^2 = 93\%$, $P < 0.001$; Fig 10).

Twenty-four studies provided a total of 25 effect sizes for parental depressive symptoms. More than half of studies (59%) reported any details regarding reliability or validity of the measure of depressive symptoms. The Center for Epidemiologic Studies Depression Scale [48] was the most commonly used measure in 2 out of 5 studies (41%). The pooled result did not indicate a significant reduction in caregiver depressive symptoms (SMD = −0.07, 95% CI: −0.16, 0.02, $P = 0.08$; $I^2 = 76\%$, $P < 0.001$; Fig 11).

## Moderator analyses

Given the large observed heterogeneity across all but one of the pooled effect size estimates, we conducted several moderator analyses to investigate potential sources of this heterogeneity. Tables 2 and 3 present the effects on ECD and parenting outcomes, respectively, stratified by the 7 characteristics: country-income level, average child age at baseline, intervention content, intervention duration, delivery, setting, and study quality. Parenting interventions had a greater effect on child cognitive, language, and motor development and parenting practices in LMICs compared to HICs, and the subgroup differences for these outcomes were statistically significant (e.g., effects on cognitive development: SMD = 0.41, 95% CI: 0.29, 0.53 in LMICs versus SMD = 0.17, 95% CI: 0.10, 0.22 in HICs; $P < 0.001$ for difference between subgroups;

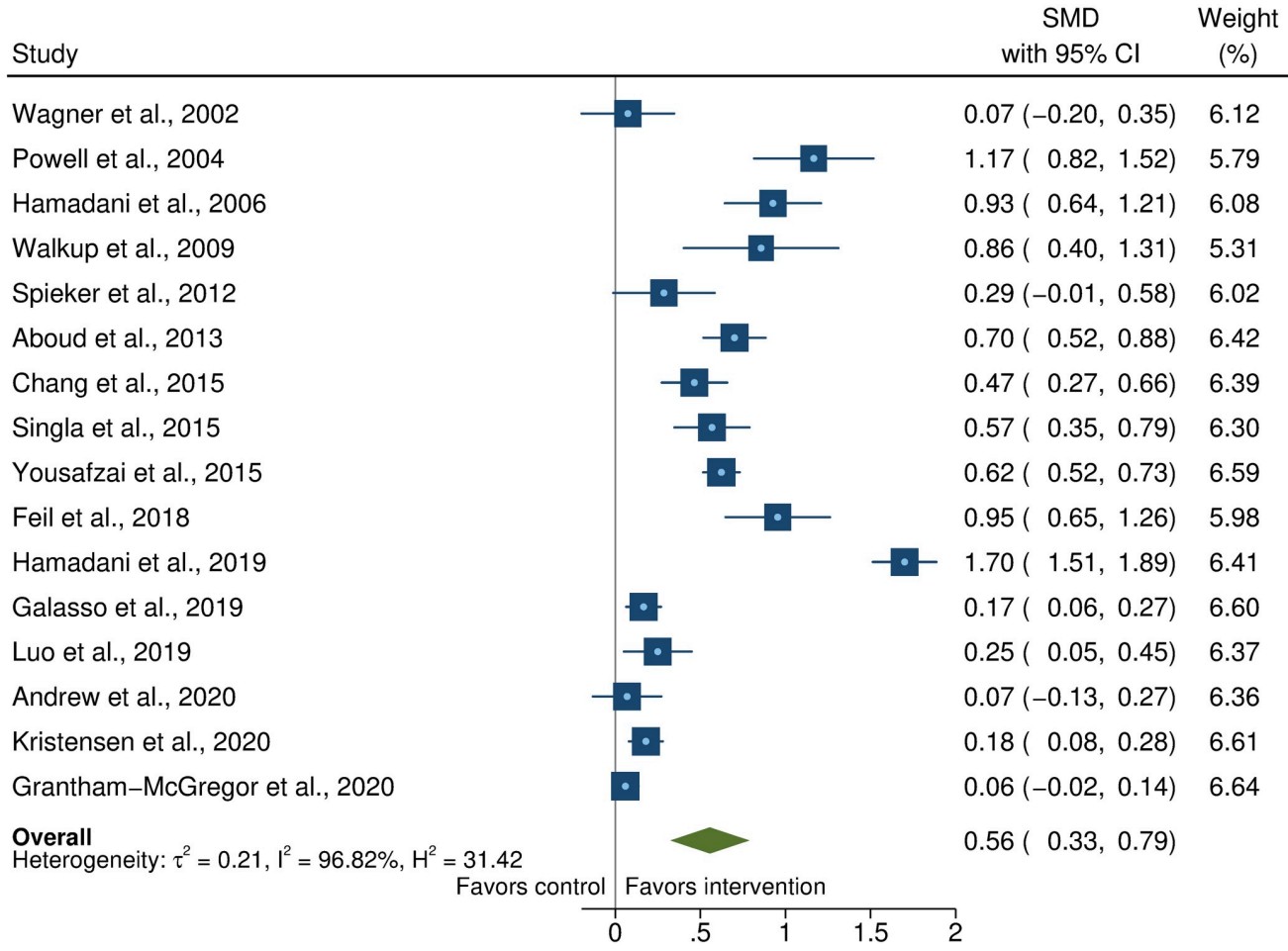

**Fig 8. Forest plot for the effect of parenting interventions on parenting knowledge.** Blue squares represent the SMD for each study, with the size of the square being proportional to the study weight. The whiskers extending from each side of the square represent the range of the 95% CI. The green diamond shows the overall pooled effect size using a random-effects model, which is centered at the point estimate and the diamond width representing the 95% CI. SMD, standardized mean difference.

Table 2). Although effects on child socioemotional development, parenting knowledge, parent–child interactions, and parental depressive symptoms did not significantly differ by country-income level, the magnitudes of the estimates were consistently greater for all outcomes in LMICs versus HICs (Table 3).

Parenting interventions that promoted responsive caregiving had significantly greater effects on child cognitive development, parenting knowledge, practices, and parent–child interactions compared to interventions without responsive caregiving (e.g., effects on parenting practices: SMD = 0.42, 95% CI: 0.29, 0.56 for interventions with responsive caregiving components versus SMD = 0.11, 95% CI: 0.01, 0.22 for those without; $P$ = 0.001 for difference between subgroups; Table 3). With the exceptions of greater subgroup effects on cognitive development for programs targeting children older than 12 months of age and greater effects on parenting practices for programs less than 12 months in duration, there were no other

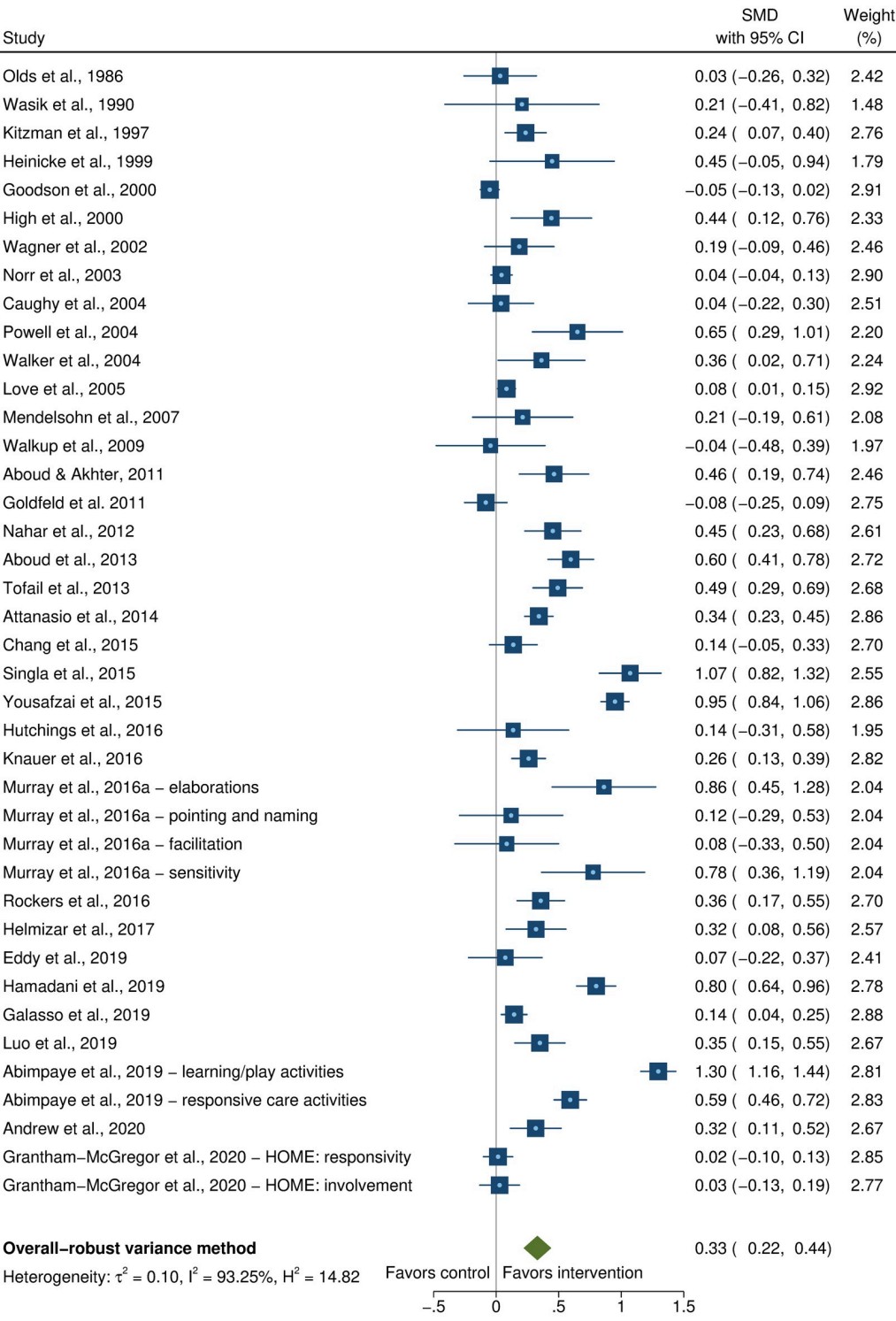

**Fig 9. Forest plot for the effect of parenting interventions on parenting practices.** Blue squares represent the SMD for each study, with the size of the square being proportional to the study weight. The whiskers extending from each side of the square represent the range of the 95% CI. The green diamond shows the overall pooled effect size using a random-effects model, which is centered at the point estimate and the diamond width representing the 95% CI. SMD, standardized mean difference.

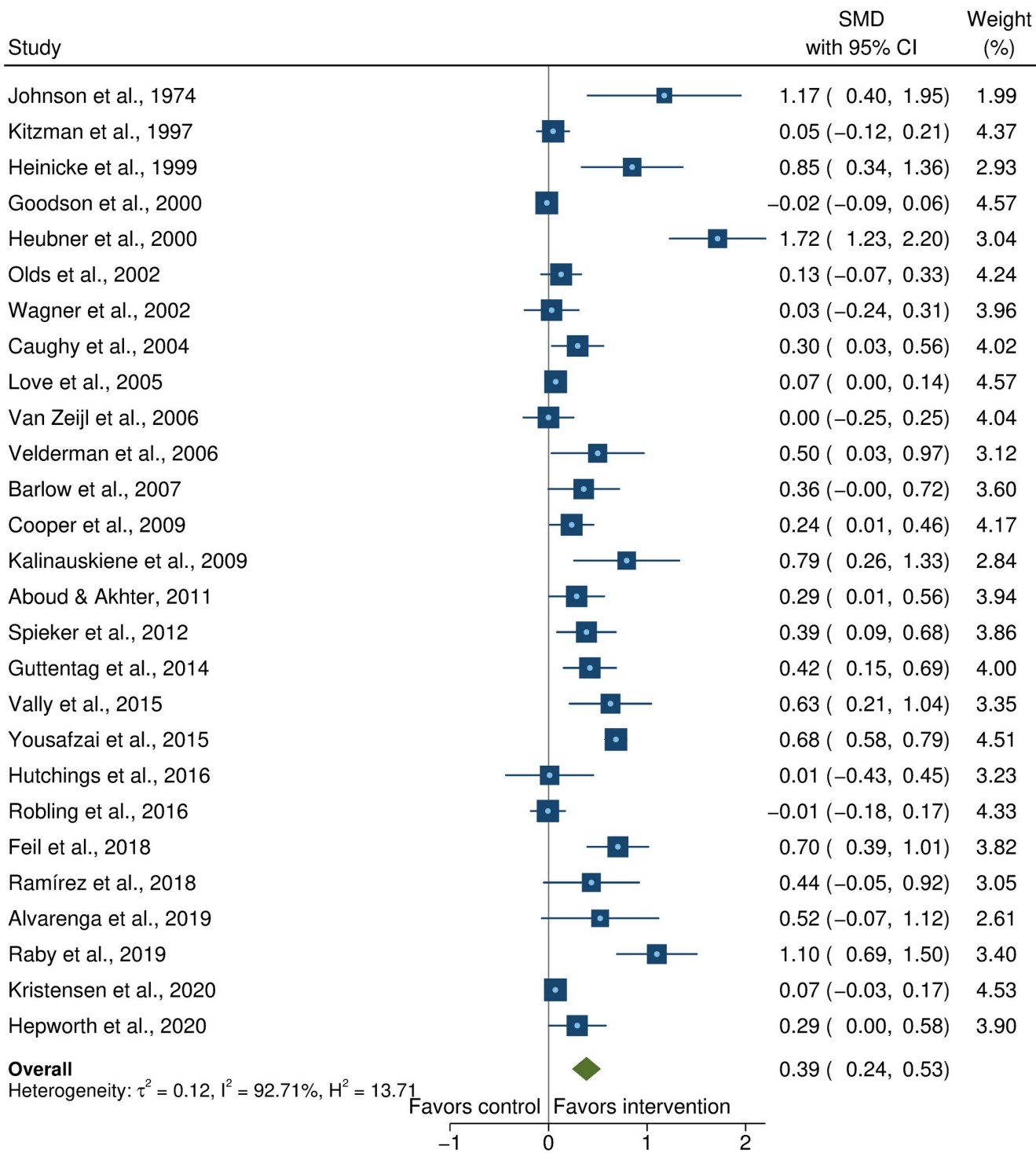

**Fig 10. Forest plot for the effect of parenting interventions on parent–child interactions.** Blue squares represent the SMD for each study, with the size of the square being proportional to the study weight. The whiskers extending from each side of the square represent the range of the 95% CI. The green diamond shows the overall pooled effect size using a random-effects model, which is centered at the point estimate and the diamond width representing the 95% CI. SMD, standardized mean difference.

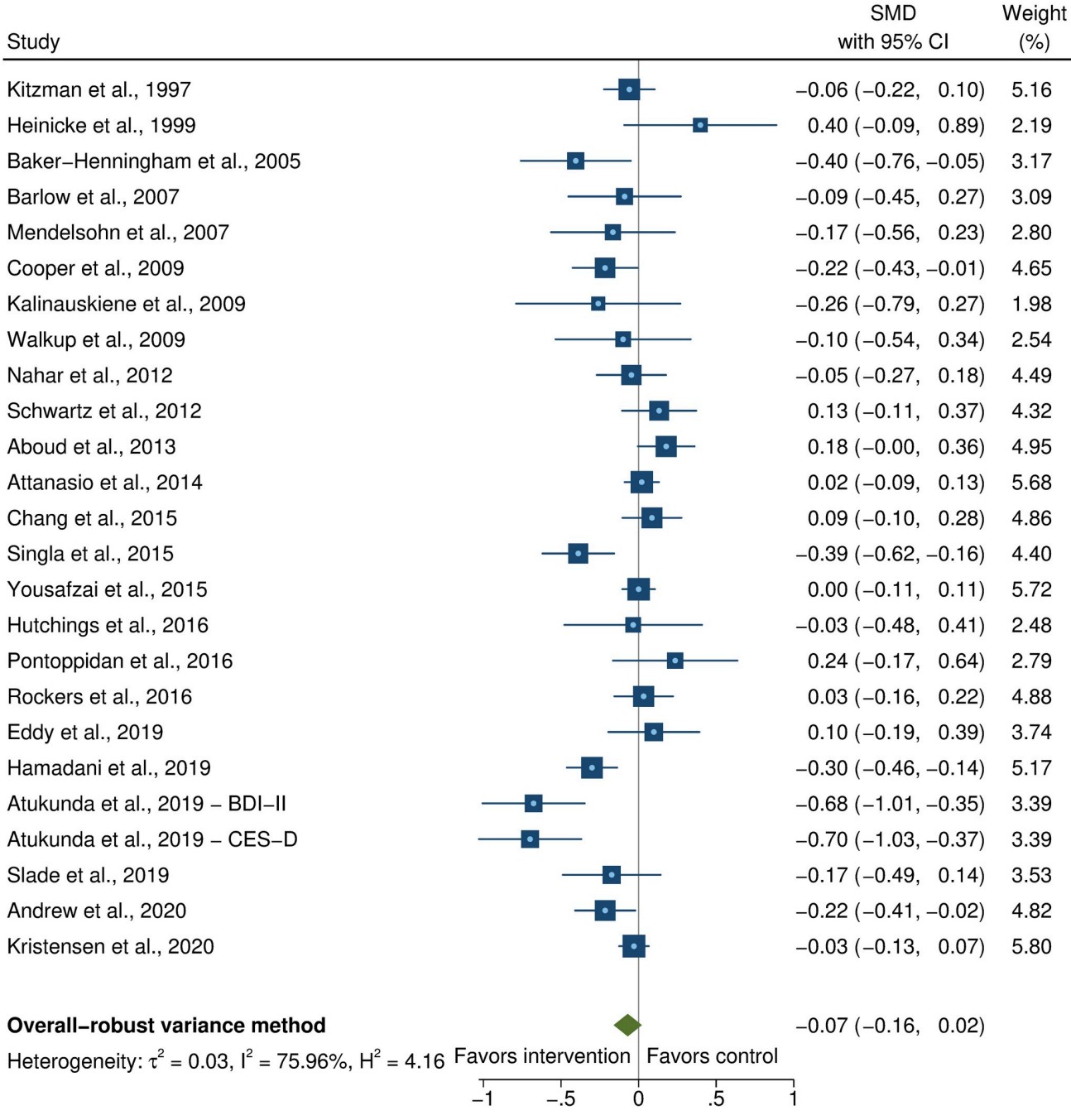

**Fig 11. Forest plot for the effect of parenting interventions on parental depressive symptoms.** Blue squares represent the SMD for each study, with the size of the square being proportional to the study weight. The whiskers extending from each side of the square represent the range of the 95% CI. The green diamond shows the overall pooled effect size using a random-effects model, which is centered at the point estimate and the diamond width representing the 95% CI. SMD, standardized mean difference.

**Table 2. Subgroup results for the effect of parenting interventions on early child development outcomes stratified by study characteristics.**

| Moderator | N | Cognitive development SMD (95% CI) | P value | N | Language development SMD (95% CI) | P value | N | Motor development SMD (95% CI) | P value | N | Socioemotional development SMD (95% CI) | P value | N | Behavioral development SMD (95% CI) | P value | N | Attachment SMD (95% CI) | P value |
|---|---|---|---|---|---|---|---|---|---|---|---|---|---|---|---|---|---|---|
| Country economic grouping | | | | | | | | | | | | | | | | | | |
| High-income countries | 26 | **0.17 (0.10, 0.22)** | **<0.001** | 21 | **0.18 (0.06, 0.29)** | **0.02** | 6 | **0.09 (−0.01, 0.19)** | **0.03** | 16 | 0.14 (0.05, 0.23) | 0.26 | - | - | - | - | - | - |
| Low- and middle-income countries | 32 | **0.41 (0.29, 0.53)** | | 25 | **0.35 (0.21, 0.48)** | | 29 | **0.26 (0.16, 0.36)** | | 13 | 0.24 (0.08, 0.40) | | - | - | - | - | - | - |
| Child age at baseline | | | | | | | | | | | | | | | | | | |
| <12 months | 37 | **0.25 (0.15, 0.34)** | **0.02** | 26 | 0.22 (0.10, 0.35) | 0.06 | 19 | 0.22 (0.13, 0.31) | 0.28 | 20 | 0.12 (0.04, 0.19) | 0.27 | 20 | −0.14 (−0.20, −0.07) | 0.70 | - | - | - |
| ≥12 months | 21 | **0.45 (0.31, 0.59)** | | 20 | 0.35 (0.21, 0.49) | | 16 | 0.26 (0.09, 0.44) | | 9 | 0.32 (0.09, 0.54) | | 9 | −0.10 (−0.25, 0.05) | | - | - | |
| Intervention content | | | | | | | | | | | | | | | | | | |
| Responsive caregiving | 36 | **0.38 (0.26, 0.49)** | **0.04** | 29 | 0.31 (0.18, 0.45) | 0.31 | 19 | 0.23 (0.08, 0.39) | 0.98 | 15 | 0.19 (0.04, 0.35) | 0.79 | 19 | −0.12 (−0.19, −0.05) | 0.66 | - | - | - |
| No responsive caregiving | 22 | **0.22 (0.13, 0.31)** | | 17 | 0.21 (0.10, 0.32) | | 16 | 0.23 (0.14, 0.32) | | 14 | 0.17 (0.07, 0.27) | | 10 | −0.17 (−0.28, −0.05) | | - | - | |
| Intervention duration | | | | | | | | | | | | | | | | | | |
| <12 months | 17 | 0.39 (0.22, 0.56) | 0.29 | 20 | 0.38 (0.20, 0.55) | 0.05 | 11 | 0.21 (0.07, 0.36) | 0.60 | 13 | 0.20 (0.09, 0.31) | 0.08 | 17 | −0.12 (−0.18, −0.07) | 0.91 | 5 | 0.21 (0.04, 0.37) | 0.21 |
| ≥12 months | 41 | 0.29 (0.20, 0.37) | | 26 | 0.23 (0.11, 0.35) | | 24 | 0.25 (0.13, 0.36) | | 16 | 0.18 (0.04, 0.32) | | 12 | −0.15 (−0.29, −0.01) | | 6 | 0.35 (0.21, 0.49) | |
| Delivery | | | | | | | | | | | | | | | | | | |
| Individual only | 30 | 0.25 (0.18, 0.32) | 0.31 | 23 | 0.23 (0.14, 0.33) | 0.45 | 18 | 0.16 (0.10, 0.22) | 0.09 | 15 | 0.12 (0.02, 0.22) | 0.10 | 17 | −0.16 (−0.22, −0.10) | 0.06 | 7 | 0.24 (0.10, 0.38) | 0.25 |
| Group only | 7 | 0.48 (0.15, 0.80) | | 8 | 0.38 (−0.03, 0.78) | | 3 | 0.71 (−0.40, 1.82) | | 5 | 0.44 (0.04, 0.83) | | 4 | −0.14 (−0.65, 0.37) | | - | - | |
| Individual and Group | 21 | 0.32 (0.17, 0.47) | | 15 | 0.27 (0.10, 0.45) | | 14 | 0.19 (0.10, 0.29) | | 9 | 0.15 (0.04, 0.26) | | 7 | −0.09 (−0.18, −0.01) | | 4 | 0.37 (0.19, 0.55) | |
| Setting | | | | | | | | | | | | | | | | | | |
| Home visiting only | 25 | 0.27 (0.18, 0.36) | 0.61 | 16 | 0.24 (0.13, 0.35) | 0.70 | 15 | 0.17 (0.09, 0.25) | 0.55 | 9 | 0.12 (−0.05, 0.28) | 0.51 | 11 | −0.13 (−0.21, −0.06) | 0.07 | 8 | 0.26 (0.13, 0.39) | 0.47 |
| Community only | 6 | 0.37 (0.19, 0.54) | | 7 | 0.25 (−0.10, 0.60) | | - | - | | - | - | | 3 | −0.15 (−1.15, 0.85) | | - | - | |
| Clinic only | 7 | 0.45 (0.14, 0.77) | | 8 | 0.27 (−0.07, 0.62) | | 6 | 0.32 (−0.18, 0.81) | | 6 | 0.30 (−0.07, 0.66) | | 5 | −0.23 (−0.34, −0.12) | | - | - | |
| Combination | 20 | 0.29 (0.12, 0.43) | | 14 | 0.29 (0.11, 0.48) | | 12 | 0.21 (0.11, 0.31) | | 12 | 0.18 (0.07, 0.28) | | 10 | −0.10 (−0.18, −0.02) | | 3 | 0.35 (0.16, 0.53) | |
| Study quality | | | | | | | | | | | | | | | | | | |

*(Continued)*

**Table 2.** (Continued)

| Moderator | Cognitive development | | | Language development | | | Motor development | | | Socioemotional development | | | Behavioral development | | | Attachment | | |
|---|---|---|---|---|---|---|---|---|---|---|---|---|---|---|---|---|---|---|
| | N | SMD (95% CI) | P value | N | SMD (95% CI) | P value | N | SMD (95% CI) | P value | N | SMD (95% CI) | P value | N | SMD (95% CI) | P value | N | SMD (95% CI) | P value |
| Lower risk of bias (<6) | 44 | 0.33 (0.23, 0.43) | 0.31 | 35 | 0.28 (0.16, 0.40) | 0.61 | 28 | 0.24 (0.13, 0.34) | 0.53 | 19 | 0.22 (0.10, 0.34) | 0.15 | 18 | −0.10 (−0.16, −0.03) | 0.05 | 6 | 0.29 (0.13, 0.44) | 0.88 |
| Higher risk of bias (≥6) | 14 | 0.26 (0.16, 0.35) | | 11 | 0.26 (0.12, 0.39) | | 7 | 0.20 (0.12, 0.29) | | 10 | 0.12 (−0.02, 0.26) | | 11 | −0.22 (−0.34, −0.08) | | 5 | 0.29 (0.18, 0.40) | |

$N$ = number of RCTs represented in subgroup analysis. SMD, standardized mean difference. $P$ value corresponds to test of subgroup differences. Bolded values indicate significant moderator effect ($P < 0.05$). Empty cells represent cases in which there were two or fewer studies for a given subgroup; if there was only 1 subgroup with a sufficient sample, then the moderator analysis was not conducted.

subgroup differences on any outcomes by child age, intervention duration, delivery, setting, or study risk of bias score (Tables 2 and 3).

## Discussion

To the best of our knowledge, this is the largest and most comprehensive systematic review and meta-analysis that synthesized 102 parenting interventions delivered during early childhood from across 33 countries and estimated the effectiveness on a wide range of ECD and caregiving outcomes. We found that parenting interventions delivered during the first 3 years of life improved early child cognitive, language, motor, and socioemotional development, and attachment, and reduced behavior problems. Parenting interventions additionally improved parenting knowledge, parenting practices, and parent–child interactions; however, they did not significantly reduce parental depressive symptoms. Our findings extend prior evidence in 4 key ways.

First, previous reviews of parenting interventions delivered during early childhood have focused on the effectiveness of select types of parenting interventions on 1 or 2 developmental outcomes [6,49]. We conducted the most comprehensive synthesis that evaluates intervention effects on 6 distinct child development outcomes that are most relevant to children during the earliest years of life. Overall, pooled effect sizes were relatively greater for cognitive, language, and motor development and attachment, but smaller for socioemotional development and behavior problems. Greater estimated pooled effects on cognitive, language, and motor development are consistent with the strategies used in the majority of reviewed parenting interventions that focused largely on increasing early play and learning opportunities—such as through caregiver engagement in stimulation activities or enhanced verbal responsiveness—which are directly related to these particular outcomes. In contrast, the smaller observed effects on socioemotional development and behavior problems are likely because fewer of the reviewed parenting interventions incorporated program components directly related to the commonly assessed outcome domains such as emotional competence or externalizing problem behaviors.

In order to effectively improve child socioemotional development and reduce behavior problems, interventions require evidence-based curriculum content and strategies that are grounded in social learning theory, such as supporting caregivers' behavioral management skills, addressing parental mental health, and encouraging nonviolent discipline [23,50]. Yet, most parenting programs for socioemotional and behavioral development to date have been

**Table 3. Subgroup results for the effect of parenting interventions on parenting outcomes stratified by study characteristics.**

| Moderator | Parenting knowledge | | | Parenting practices | | | Parent–child interactions | | | Parental depressive symptoms | | |
|---|---|---|---|---|---|---|---|---|---|---|---|---|
| | N | SMD (95% CI) | P value | N | SMD (95% CI) | P value | N | SMD (95% CI) | P value | N | SMD (95% CI) | P value |
| Country economic grouping | | | | | | | | | | | | |
| High-income countries | 5 | 0.45 (0.10, 0.80) | 0.51 | **15** | **0.08 (−0.01, 0.16)** | **<0.001** | 22 | 0.38 (0.20, 0.55) | 0.52 | 12 | −0.02 (−0.09, 0.05) | 0.13 |
| Low- and middle-income countries | 11 | 0.60 (0.30, 0.90) | | **20** | **0.47 (0.34, 0.61)** | | 5 | 0.47 (0.25, 0.69) | | 12 | −0.13 (−0.27, 0.01) | |
| Child age at baseline | | | | | | | | | | | | |
| <12 months | 8 | 0.48 (0.26, 0.70) | 0.55 | 21 | 0.23 (0.11, 0.35) | 0.27 | 18 | 0.29 (0.16, 0.42) | 0.14 | 16 | −0.03 (−0.13, 0.07) | 0.47 |
| ≥12 months | 8 | 0.62 (0.21, 1.04) | | 14 | 0.47 (0.29, 0.66) | | 9 | 0.59 (0.22, 0.96) | | 8 | −0.16 (−0.32, 0.01) | |
| Intervention content | | | | | | | | | | | | |
| Responsive caregiving | **11** | **0.68 (0.39, 0.97)** | **0.001** | **24** | **0.42 (0.29, 0.56)** | **0.001** | **22** | **0.47 (0.30, 0.64)** | **<0.001** | 18 | −0.08 (−0.18, 0.01) | 0.82 |
| No responsive caregiving | **5** | **0.19 (0.13, 0.26)** | | **11** | **0.11 (0.01, 0.22)** | | **5** | **0.05 (−0.01, 0.12)** | | 6 | −0.07 (−0.34, 0.21) | |
| Intervention duration | | | | | | | | | | | | |
| <12 months | 6 | 0.57 (0.32, 0.82) | 0.92 | **11** | **0.51 (0.30, 0.72)** | **0.01** | 14 | 0.44 (0.23, 0.65) | 0.05 | 10 | −0.13 (−0.32, 0.07) | 0.27 |
| ≥12 months | 10 | 0.54 (0.20, 0.89) | | **24** | **0.25 (0.13, 0.37)** | | 13 | 0.33 (0.13, 0.53) | | 14 | −0.05 (−0.14, 0.04) | |
| Delivery | | | | | | | | | | | | |
| Individual only | 8 | 0.46 (0.18, 0.75) | 0.92 | 15 | 0.24 (0.12, 0.35) | 0.14 | 15 | 0.32 (0.17, 0.48) | 0.67 | 12 | −0.08 (−0.17, 0.01) | 0.14 |
| Group only | - | - | | 6 | 0.42 (0.16, 0.68) | | 4 | 0.65 (−0.07, 1.37) | | 4 | −0.22 (−0.83, 0.39) | |
| Individual and Group | 7 | 0.48 (0.25, 0.72) | | 14 | 0.38 (0.15, 0.60) | | 8 | 0.36 (0.10, 0.62) | | 8 | 0.02 (−0.13, 0.17) | |
| Setting | | | | | | | | | | | | |
| Home visiting only | 7 | 0.39 (0.11, 0.68) | 0.64 | 12 | 0.27 (0.15, 0.38) | 0.65 | 16 | 0.35 (0.19, 0.51) | 0.50 | 12 | −0.07 (−0.17, 0.02) | 0.67 |
| Community only | - | - | | 5 | 0.32 (0.16, 0.48) | | 4 | 0.65 (−0.07, 1.37) | | 3 | −0.17 (−1.39, 1.04) | |
| Clinic only | - | - | | 6 | 0.33 (−0.01, 0.67) | | - | - | | 4 | −0.10 (−0.43, 0.22) | |
| Combination | 6 | 0.49 (0.21, 0.77) | | 12 | 0.39 (0.11, 0.65) | | 6 | 0.24 (0.00, 0.48) | | 5 | −0.01 (−0.27, 0.24) | |
| Study quality | | | | | | | | | | | | |
| Lower risk of bias (<6) | 13 | 0.60 (0.34, 0.87) | 0.48 | 27 | 0.37 (0.25, 0.50) | 0.06 | 18 | 0.36 (0.22, 0.51) | 0.08 | 19 | −0.10 (−0.20, 0.01) | 0.39 |
| Higher risk of bias (≥6) | 3 | 0.33 (−0.10, 0.76) | | 8 | 0.12 (−0.04, 0.28) | | 9 | 0.47 (0.09, 0.85) | | 5 | −0.02 (−0.24, 0.20) | |

$N$ = number of RCTs represented in subgroup analysis. SMD, standardized mean difference. $P$ value corresponds to test of subgroup differences. Bolded values indicate significant moderator effects ($P < 0.05$). Empty cells represent cases in which there were two or fewer studies for a given subgroup.

commonly delivered for preschool-aged children rather than young children during the earliest years of life [23]. New intervention models are needed in order to more effectively improve early child socioemotional and behavioral development particularly for young children during infancy and early childhood. Part of this effort will require establishing standardized measurement tools for assessing socioemotional development and behavioral problems that can be used reliably and cross-culturally in the context of interventions with young children [51].

Second, we analyzed the impact of parenting interventions on parents themselves and found improvements in parenting knowledge, parenting practices, and parent–child interactions. However, 40% of studies aimed at improving ECD did not measure a single parenting outcome. Given that the primary focus of parenting interventions is to support parents and improve the caregiving environment, it is critical to measure parenting as a way of understanding how interventions impact ECD [21,52]. More mediation analyses are needed to empirically substantiate program theories of change and identify the key caregiving mechanisms by which parenting programs lead to improved ECD outcomes [36,53–55].

We found little evidence to support that parenting interventions reduced parental depressive symptoms. By covering a broader set of parenting interventions and including studies in HICs, our results extend the findings of a prior meta-analysis that found that stimulation

interventions did not significantly reduce maternal depressive symptoms in LMICs [21]. This overall null effect on depressive symptoms is likely due to the fact that few of the parenting interventions had an explicit psychological component to address parental mental health. However, one notable exception was a parenting group intervention in Uganda that integrated sessions on not only stimulation and responsive caregiving but also parental emotion regulation, coping strategies, and principles of cognitive behavioral therapy. This intervention found medium-sized reductions in maternal depression [36]. While bundling multiple components together (e.g., mental health and parenting) to create intervention packages is recommended in the literature [5], additional research is needed to determine the independent and additive benefits of particular components (e.g., those that target mental health versus parenting) in order to optimize intervention effects on outcomes.

In spite of the observational evidence that fathers are critically important for ECD [56], only 7 parenting interventions in this review (7%) engaged fathers to some degree [34–40], and only 1 study measured paternal outcomes from fathers directly (who only comprised 5% of the respondents in this study) [39]. A prior review has drawn attention to the limited number of parenting interventions with fathers and the program implementation barriers that limit fathers' participation in parenting interventions across global contexts [57]. More research is needed to determine how to optimize the design and delivery of parenting programs so that they are inclusive of fathers and can achieve benefits for maternal and paternal parenting and ECD [58].

Third, moderator analyses showed that parenting interventions had greater effects in LMICs compared to HICs for improving child cognitive, language, and motor development and parenting practices. Parents and children in LMICs are more likely to be exposed to additional risk factors that constrain nurturing care and ECD (e.g., low parental education, malnutrition, fewer early learning opportunities) [3,59], such that support for parenting may be more beneficial in low-resource contexts. More research is needed to understand whether and how variation in risk profiles within populations may influence the degree of benefits from a parenting intervention [60]. It is worth noting that none of the included studies directly compared the same intervention program in a LMIC and HIC. As a result, it is possible that these subgroup results may also, in part, be confounded by other differences in intervention design and implementation between LMICs and HICs.

Regarding intervention content, we found that those with responsive caregiving components had significantly greater impacts on child cognitive development, parenting practices, and parent–child interactions, compared to interventions that did not include responsive caregiving content. These results underscore the importance of not only increasing early play and learning materials (e.g., provision of homemade toys), which have traditionally been the focus of parenting intervention for young children especially in LMICs, but also incorporating program components that directly support parental sensitivity and responsiveness [25,61].

At the same time, we did not find evidence to suggest that child age, intervention duration, delivery modality, or setting moderated intervention effects on more than one outcome. The lack of statistically significant subgroup results by child age are consistent with the few prior meta-analyses that have similarly shown no age effects among parenting interventions focused during early childhood [9,22] or spanning broader age ranges from early to middle childhood [49] or birth to age 18 [62]. Despite strong neuroscience and economic arguments for intervening as early in life as possible [4,63], our review suggests that parental engagement and support programs may indeed have universal benefits on child and parent outcomes irrespective of child age or the timing of introduction.

Furthermore, the lack of statistically significant subgroup results by program duration, delivery modality, and setting are also consistent with the few prior meta-analyses that have

conducted similar analyses and shown no associations between these program aspects and intervention effects on ECD and parent-level outcomes [62]. One recently published cluster RCT specifically compared the effectiveness of weekly home visits to weekly mother–child group sessions over a 2-year period in rural India [64]. Trial results revealed that group sessions were as effective as home visits in improving child cognition and language, and both delivery models similarly had null impacts on child motor, socioemotional, behavior development, and parenting knowledge and practices. Of note, cost analysis revealed that the group-based model required a quarter of the costs compared to home visiting. Nevertheless, our findings suggest that there are various effective implementation strategies and program models that can be used to deliver parenting interventions and achieve positive impacts on ECD and parent outcomes. Decisions regarding program duration, delivery modality, and setting should be determined based on the existing resources and systems, community needs, risk profiles of the population, and cultural context.

In summary, our study had several strengths. These include the comprehensive review of RCTs conducted in LMICs and HICs, the investigation of multiple child development and caregiver outcomes, the large number of studies and participants represented, and the various moderator analyses conducted to explore potential features associated with intervention effectiveness. At the same time, there are a number of limitations that should be noted. First, there was considerable unexplained heterogeneity in the pooled effects across outcomes. Future research should further explore heterogeneity in treatment effects by other program factors (e.g., content, theoretical goals, behavior change techniques) and intervention implementation features (e.g., dosage, fidelity, population risk profiles, supervision and training provided to delivery agents). Considering the generally inadequate reporting of intervention details that we identified across the majority of studies, future trials should use standardized guidelines such as CARE [65] and TIDieR [66] for common reporting of intervention content and implementation to facilitate more nuanced comparisons and subgroup analyses. Second, our review focused on parenting interventions for children during the first 3 years of life. As a result, other relevant parenting interventions with preschool-aged children [67,68] or those covering a broader age range of children (e.g., 2- to 6-year-olds) [69,70] were excluded, which limits the generalizability of our results. Third, the reporting of the psychometric properties of measures (e.g., reliability and validity evidence, adaptation procedures) was highly variable across studies, with many studies not reporting any such details. Measures of ECD and parenting ranged widely from validated and adapted assessments to unstandardized measures, which may affect comparability across studies and robustness of findings. Fourth, our meta-analysis involved 6 ECD outcomes and 4 parent-level outcomes and 7 moderator variables, which raises the issue of statistical multiplicity and the increased risk for false positive findings in our estimates of pooled effect sizes and subgroup analyses. Although we used robust variance estimation to address multiplicity and statistical dependencies, this method might raise issues of disproportionate influence from studies that contribute a large number of effect sizes relative to those that contribute fewer, and particularly when the overall number of studies and effect sizes is relatively small [71]. Future reviews should apply additional methods to address issues of multiplicity. Fifth, we found evidence suggesting small-sample bias for 2 outcomes (child language and parent–child interactions) based on the results of Egger's tests. Future reviews should include searches of grey literature, regional databases, and in multiple languages to improve the global coverage and reduce publication bias in the evidence base.

Finally, our review also highlighted other broader issues with regard to risk of bias in the design, conduct, and reporting of parenting intervention trials. Specifically, the majority of trials had potential concerns of selection bias due to inadequate details about what methods were used to conceal the allocation sequence before assignment to determine whether intervention

allocations could have been foreseen before or during enrolment. Additionally, the majority of trials were not preregistered and did not have a study protocol of prespecified outcomes to assess whether selective outcome reporting bias or selective analysis reporting bias was present. Selective reporting bias, especially of null results, may additionally contribute to the publication bias detected for some of the outcomes. In light of our findings, future trials of parenting interventions should preregister primary and secondary outcomes and their scoring methods in study protocols, report all analyzed outcomes regardless of statistical significance levels, and provide adequate descriptions in publications not only regarding random sequence generation but also the allocation concealment process to especially improve the quality of this body of evidence.

## Conclusions

Parenting interventions during the first 3 years of childhood are effective for improving ECD and caregiving. In the last decade, there has been considerable policy attention on ECD globally [8]. The United Nation's Sustainable Development Goal Target 4.2 is to ensure that all girls and boys have access to quality early childhood development, care, and education services [72]. However, the current policy landscape for ECD has not yet resulted in greater investments and implementation of large-scale national parenting programs. More implementation science research regarding how to adapt and deliver parenting programs across diverse local contexts, coordinate within existing systems and services, finance programs so that they are cost-effective and sustainable, and scale parenting programs while maintaining quality and effectiveness is needed to accelerate political will and action. Finally, while the present review supports robust immediate benefits of parenting interventions during early childhood, more follow-up studies of individual trials are needed to determine whether there are sustained, longer-term effects for children and caregivers over the life course [73]. Greater knowledge regarding the sustainability or fadeout of effects can guide the design and packaging of improved parenting programs, such as the integration of subsequent "booster" sessions that can potentially further support children and caregivers' evolving needs and sustain intervention benefits over the life course.

## Supporting information

**S1 PRISMA Checklist.**
(DOC)

**S1 Protocol. Final review protocol.**
(DOCX)

**S1 Text. Search strategy.**
(DOCX)

**S1 Table. Definition of parenting interventions and those that focus on responsive caregiving versus those that do not.**
(DOCX)

**S2 Table. Measures used for each ECD and parenting outcome domain.**
(DOCX)

**S1 Fig. Risk of bias across all randomized controlled trials.**
(DOCX)

## Acknowledgments

We thank Carol Mita, research librarian at the Francis A. Countway Library of Medicine at the Harvard Longwood Campus, for her assistance in the literature search. We thank the following graduate research assistants for their contributions to data extraction during an early stage of the study: Alastair Fung, Raghbir Kaur, Helen Pitchik, Vijayaragavan Prabakaran, Mathilda Regan, Katharine Robb, Rucha Shelgikar, and Yu-Cheng Tsai. We thank the members of the WHO Guideline Development Group for their contributions during an early phase of this evidence review. We also acknowledge the support of Bernadette Daelmans, Tarun Dua, and Nigel Rollins at the World Health Organization.

## Author Contributions

**Conceptualization:** Joshua Jeong, Emily E. Franchett, Aisha K. Yousafzai.

**Data curation:** Joshua Jeong, Emily E. Franchett, Clariana V. Ramos de Oliveira, Karima Rehmani.

**Formal analysis:** Joshua Jeong.

**Funding acquisition:** Aisha K. Yousafzai.

**Investigation:** Joshua Jeong, Emily E. Franchett.

**Methodology:** Joshua Jeong.

**Project administration:** Joshua Jeong, Emily E. Franchett.

**Software:** Joshua Jeong.

**Supervision:** Joshua Jeong, Aisha K. Yousafzai.

**Visualization:** Joshua Jeong.

**Writing – original draft:** Joshua Jeong.

**Writing – review & editing:** Joshua Jeong, Emily E. Franchett, Clariana V. Ramos de Oliveira, Karima Rehmani, Aisha K. Yousafzai.

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
