## [Editor Report · Decision Letter 0]

5 Oct 2020

Dear Dr Jeong, 

Thank you for submitting your manuscript entitled "Parenting Interventions to Promote Early Child Development in the First Three Years of Life: A Systematic Review and Meta-Analysis" for consideration by PLOS Medicine.

Your manuscript has now been evaluated by the PLOS Medicine editorial staff, as well as by an academic editor with relevant expertise, and I am writing to let you know that we would like to send your submission out for external peer review.

Kind regards,

Caitlin Moyer, Ph.D.,

Associate Editor

PLOS Medicine

---

## [Decision Letter · Decision Letter 1]

2 Nov 2020

Dear Dr. Jeong,

Thank you very much for submitting your manuscript "Parenting Interventions to Promote Early Child Development in the First Three Years of Life: A Systematic Review and Meta-Analysis" (PMEDICINE-D-20-04650R1) for consideration at PLOS Medicine. 

[LINK]

In light of these reviews, I am afraid that we will not be able to accept the manuscript for publication in the journal in its current form, but we would like to consider a revised version that addresses the reviewers' and editors' comments. Obviously we cannot make any decision about publication until we have seen the revised manuscript and your response, and we plan to seek re-review by one or more of the reviewers. 

We expect to receive your revised manuscript by Nov 23 2020 11:59PM. Please email us (plosmedicine@plos.org) if you have any questions or concerns.

We look forward to receiving your revised manuscript. 

Sincerely,

Emma Veitch, PhD

PLOS Medicine

On behalf of Caitlin Moyer, PhD, Associate Editor, 

PLOS Medicine

plosmedicine.org

*Regarding the paper format/style - please structure your abstract using the PLOS Medicine headings (Background, Methods and Findings, Conclusions) -writeup in each section should use full sentences rather than bullet-point type sentence fragments. In the last sentence of the Abstract Methods and Findings section, please include a brief note about any key limitation(s) of the study's methodology. Finally please also head up the first section of the paper as "Introduction". 

*At this stage, we ask that you include a short, non-technical Author Summary of your research to make findings accessible to a wide audience that includes both scientists and non-scientists. The Author Summary should immediately follow the Abstract in your revised manuscript. This text is subject to editorial change and should be distinct from the scientific abstract. Please see our author guidelines for more information: https://journals.plos.org/plosmedicine/s/revising-your-manuscript#loc-author-summary

*If possible please reformat the citation style into PLOS Medicine's format (should be straight forward if using referencing software) - this should use callouts formatted as sequential numerals in square brackets (not superscript).

*The reviewers have noted that the date for search cutoff was May 2019, which now makes the systematic review a little dated. The editors felt that updating the search to the present would be ideal in order to address this concern.

*We noted that although the authors state the paper was reported with the use of the PRISMA reporting tool, we did not see a completed PRISMA checklist in the supplementary files - please download the word or pdf version of the checklist at http://www.prisma-statement.org/PRISMAStatement/Checklist.aspx. When completing the checklist (which should be included as part of the supplementary materials), please use section and paragraph numbers, rather than page numbers.

*In addition reviewers noted that an omission in the systematic review methods was the lack of consideration of publication bias (or other possible forms of bias affecting the array of effects across studies), please note that a component of PRISMA is the assessment of risk of bias across studies (items 15 and 22). See the PRISMA paper (elaboration and explanation) at https://journals.plos.org/plosmedicine/article?id=10.1371/journal.pmed.1000100#s6 which sets out what can be reported and analysed under those points. 

Comments from the reviewers:

Reviewer #1: This paper describes the carrying out of a systematic review and meta-analysis of parenting interventions to promote early child development in the first three years of life.

This is an important review in bringing together in one place a large body of evidence. It will therefore be a helpful reference in the future to locate research in this field.

I have a few comments that may help to improve the usefulness of this paper.

Abstract

The outcomes included have a list that includes "…behaviour development..". Should this read "..behaviour, development.." (i.e. is there a missing comma?) or does this refer to behaviour and the development of behaviour. Please clarify. Also check that this list is consistent with later repeated version of the list which subtly differ. Please check and amend as necessary.

Introduction

The introduction is important for readers to build up a sense of the literature on the subject.

The sentence in para 2 of the introduction beginning "Parenting interventions can encompass a range of interventions including…." will need improved referencing. I would suggest using systematic reviews or theoretical papers to lay out the types of parenting interventions available. 

Also there are some big categories missing such as parental mental health, interventions to develop parental sensitivity etc. Please check the literature and make sure you include the main categories so the reader can orientate to the review.

In para 3 the authors say that a review published in 2018 is "relatively outdated". I hope that we can still draw some information from a review published in the last ten years, so I would perhaps remove this or use different terminology. Perhaps keep to the focus on what you have done differently, which is clear.

Methods

I would usually expect to see criteria laid out in the usual PICOS criteria way. Is there a reason you have not done this? 

I think it would have been stronger if you had not excluded children with disability or parents with disability. Please discuss this limitation in the relevant section.

Based on your exclusion criteria it would seem that if an intervention targeted 0-7 year old children it would be excluded as the average age would be 3.5 years. Is this correct? If so please discuss this as a limitation. This may rule out some long term interventions for example.

Were the graduate research assistants trained (please state)? Please discuss this. Is this a limitation? For example, using multiple graduates could lead to some bias. 

Results

Could you say something about what proportion of studies used validated outcome measures? (As opposed to observation/frequency counts/bespoke measures etc.). This may be important information for future researchers. 

Discussion

You have highlighted shortcomings of studies in terms of quality and bias. It would be helpful to have a very brief paragraph outlining the particular aspects future studies need to do/improve upon to generate better evidence. You have done this to an extent. I think this is very important aspect of this paper.

In the discussion you say that effects in LMICs are larger than in HICs.Can you clarify if any studies have directly compared the same intervention programme in those two settings or is it the case that the different studies use different interventions but show larger effect sizes. If the latter can you draw this conclusion? The different intervention used could be a factor for example.

The limitations section is well written although please include the things I have mentioned above for thoroughness. 

In several of the areas mentioned there are existing systematic reviews that I would have expected to be referenced as the main systematic reviews in place prior to this one. 

These for example include:

Behaviour

Moon, D. J., Damman, J. L., & Romero, A. (2020). The effects of primary care-based parenting interventions on parenting and child behavioral outcomes: A systematic review. Trauma, Violence, & Abuse, 21(4), 706-724.

Attachment

Wright, B., Hackney, L., Hughes, E., Barry, M., Glaser, D., Prior, V., ... & Garside, M. (2017). Decreasing rates of disorganised attachment in infants and young children, who are at risk of developing, or who already have disorganised attachment. A systematic review and meta-analysis of early parenting interventions. Plos one, 12(7), e0180858.

etc. I think these should be in an introductory paragraph to set the context of this review so that the reader can look at the existing evidence. Here you can include these existing systematic reviews and reference them. This sets the scene for what is known and what this review adds, and helps people from different fields navigate the literature. This would greatly enhance the paper.

There are some big pieces of work that address some of the issues you mention such as a lack of involvement of fathers in parental interventions that I might have expected to see as part of the critical analysis of findings 

(e.g. Panter-Brick, C., Burgess, A., Eggerman, M., McAllister, F., Pruett, K., & Leckman, J. F. (2014). Practitioner review: engaging fathers-recommendations for a game change in parenting interventions based on a systematic review of the global evidence. Journal of child psychology and psychiatry, 55(11), 1187-1212.)

Summary

I think that this is an important piece of work and like the authors I believe we need to have improved evidence and improved research. This will hopefully set the stall out for what needs to come next.

I would suggest that this is a very publishable piece of research and hope that the authors will make suggested changes to improve it further. 

Reviewer #2: This study seeks to synthesize findings from RCTs of parenting interventions designed to promote early child development in both high income countries and low and middle income countries through a meta-analysis. A strong rationale is provided, methods are clear, and the manuscript is very well written. Findings are compelling and will inform policy, program development, and ongoing research. As such, the study has the potential to have high impact.

I have two minor concerns:

1. I would like to see more discussion about program setting, given the heterogeneity across studies (e.g., home vising, health care), and discussion of individual vs. group-based models if possible.

2. I am aware of at least two manuscripts not included that are nonetheless are highly relevant, and whose omission in each case represents a substantial limitation of the analysis:

a. Pamela C. High, Linda LaGasse, Samuel Becker, Ingrid Ahlgren and Adrian Gardner. Literacy Promotion in Primary Care Pediatrics: Can We Make a Difference? Pediatrics April 2000, 105 (Supplement 3) 927-934. While I recognize that this is described as an alternate day randomization, the biases in the design are minimal, and the study documented important impacts of the Reach Out and Read model, which is otherwise not described in this manuscript. I would also wonder if randomization processes were sufficiently described in the articles that were included, and would suggest that this be specified/defined in the methods (i.e., study exclusion criteria page 6 - "not evaluated using a randomized controlled study design").

b. Adriana Weisleder, Denise S.R. Mazzuchelli, Aline Sá Lopez, Walfrido Duarte Neto, Carolyn Brockmeyer Cates, Hosana Alves Gonçalves, Rochele Paz Fonseca, João Oliveira and Alan L. Mendelsohn. Reading Aloud and Child Development: A Cluster-Randomized Trial in Brazil. Pediatrics January 2018, 141 (1) e20170723; DOI: https://doi.org/10.1542/peds.2017-0723. This study is a cluster RCT in a low and middle income country that documents large impacts of a group-based parenting intervention in a community setting, and it appears to have not been included because the mean age of the 2-3 year olds in the study was 37 months (vs. 36) - because it is a large study, it's omission is a significant limitation.

At a minimum, I would recommend that the discussion note the exclusion of these important studies (and cite them), and perhaps note their exclusion as a limitation of the design and analyses.

These minor concerns aside, this is a very important addition to the literature.

Reviewer #3: Alex McConnachie

This is a review of the statistical aspects of the paper by Jeong et al, a systematic review and meta-analysis of parenting interventions in the first three years of life.

Overall, I found the paper to be easy to follow and interesting. Statistically, the methods that have been used appear appropriate, and are nicely presented. My comments are, in the main, relatively minor.

I note that there were 6 primary outcome domains. This multiplicity does not seem to have been taken into account in the interpretation of the results. In addition, would there have been any advantage to employing a multivariate approach, analysing primary outcomes from all six domains within the same model, allowing for correlations between intervention effects with respect to the different outcomes? This may have advantages in terms of improved estimation of pooled intervention effects, though I am certainly not an expert on these methods. The authors have done this to some extent by modelling multiple measures per domain from the same study; perhaps the potential advantages of a full multivariate analysis would not be worth the added complexity.

On page 7, I found the phrase "studies weighted by the inverse variance method (Hedge's g)" slightly confusing. I thought Hedge's g was a way to estimate the standardised mean difference, rather than to do with the inverse variance method.

For the moderator analyses, was calendar date considered as a potential moderator - i.e. have effect sizes changed over time? These studies cover a 45 year period, which seems a long time, and time enough for interventions to be improved.

One thing I thought was missing is any assessment of publication bias. Surely the authors considered this?

When reporting each outcome, in the text, there is no mention of heterogeneity, which is noticeably large for some outcomes when reviewing the figures. Also, it is not clear how many studies include multiple outcomes measuring the same domain. Perhaps we could be told the number of studies included in each analysis, and the number of effect estimates used across those studies.

In the moderator analyses, pooled effect sizes are reported within subgroups, and described as significant, but with no p-value that I could find to support such statements. For example, on page 13, "subgroup differences for these outcomes were statistically significant (e.g., for cognitive development, SMD=0.48, 95% CI: 0.33 to 0.65 in LMICs versus SMD=0.17, 95% CI: 0.09 to 0.24)" - this shows the pooled estimates within LMICs and HICs, with 95% CIs, but there is no p-value for the difference between these two estimates. It is true that these CIs do not overlap, but that need not be the case for the difference between the subgroups to be statistically significant. This is also the case in Table 2 - we have pooled effect estimates within subgroups, but no test of differences between subgroups. As an aside, I note that Table 2 reports the LMIC estimate as 0.49, not 0.48; I suggest a thorough check of the values reported in the text versus the tables.

Table 1 includes a lot of useful information, but does not give any indication of how many outcome measures each study provided in relation to each outcome domain. Rather than "X", could the table show the number of outcomes for each study within each domain?

Returning to Table 2, for some moderators, the effect estimates relate to defined subgroups of studies, but for continuous moderators, it is not clear what the estimates and confidence intervals mean. I assume these are the estimated changes in effect estimates associated with a one-unit increase in the moderator, but I would suggest this is not very helpful. It may be better, for the table, to report effect estimates within subgroups of age and intervention duration. The statistical significance of the moderating effects of continuous measures could still be reported from a meta-regression using the continuous measure.

[LINK]

---

## [Decision Letter · Decision Letter 2]

12 Mar 2021

Dear Dr. Jeong,

Thank you very much for re-submitting your manuscript "Parenting Interventions to Promote Early Child Development in the First Three Years of Life: A Global Systematic Review and Meta-Analysis" (PMEDICINE-D-20-04650R2) for review by PLOS Medicine.

I have discussed the revised paper with my colleagues and the academic editor and it was also seen again by 3 reviewers. I am pleased to say that provided the remaining editorial and production issues are dealt with we are planning to accept the paper for publication in the journal.

[LINK]

We look forward to receiving the revised manuscript by Mar 19 2021 11:59PM.   

Sincerely,

Caitlin Moyer, PhD

Associate Editor 

PLOS Medicine

plosmedicine.org

Requests from Editors:

1. Abstract: Methods and Findings: Please include p-values in addition to 95% CIs where applicable for all main results presented in the Abstract.

2. In-text citations: Throughout the text, please do not include spaces within brackets for references, for example please use [2,3] rather than [2, 3].

3. Methods: Thank you for indicating that your review was preregistered with PROSPERO and providing the information. You note two registrations associated with the review, it would be helpful if you could include as a supporting information file the final review protocol used, noting any changes from the prospectively registered protocol.

4. Results: Page 14: Please clarify if the pooled result for infant-caregiver attachment was significant “The pooled result showed a moderate positive impact on improving attachment (SMD=0.29, 95% CI: 0.18, 0.40; I2=0%, P=0.92; Fig 7).”

5. Results: Page 16: Please clarify if the pooled result for parental depressive symptoms was statistically significant: “The pooled result did not indicate a significant reduction in caregiver depressive symptoms (SMD=-0.07, 95% CI: -0.16, 0.02; I2=76%, P<0.001; Fig 11).”

6. Results: Page 17: Please clarify as moderator analyses were not done for behavioral development or attachment: “Although moderator results for the other outcomes were not significantly different by country income level, the magnitudes of the effects were all consistently greater in LMICs versus HICs.”

7. Results: Page 17: Please clarify these results are also presented in Tables 2 and 3: “However, there were no other consistent differences in outcomes by child age, intervention duration, delivery, setting, or study risk of bias score.”

8. Discussion, Page 19: Please add a space between the word and reference bracket in this sentence: “While bundling multiple components together (e.g., mental health and parenting) to create intervention packages is recommended in the literature [5]...”

9. Discussion, Page 21: Please revise to: “Our statistically non-significant moderation results by child age are consistent…” and “Furthermore, our statistically non-significant moderation results by program duration, delivery modality, and setting are also consistent with…”

or similar to clarify “non-significant” as “not statistically significant” in these sentences.

10. Table 2 and Table 3: Please define the abbreviation (SMD) in the legend.

11. Page 42: Please remove the Conflicts of Interest and Funding sections, and ensure all information is accurately entered in the “Competing Interests” and “Financial Disclosure” section of the manuscript submission form as this information would be published alongside the article.

12: Page 43: Please remove the “Human Participant Protection” statement and ensure that all relevant information pertaining to the ethical approval of your study is included in the Methods.

13. References: Please double check the consistency of all references in the list, including abbreviations and capitalizations for journal names. For example, “The Lancet Global Health” and “The Journal of nutrition” should be “Lancet Glob Health” and “J Nutr” and please check all throughout. Please similarly format the references listed in S1 Table.

14. Figures 2-11: Please give a descriptive legend to describe each figure, including all symbols, bars, and values, and including an X axis label.. Please provide the p value associated with the SMD and 95% CIs at least for the overall effects reported.

15: S1 Table: Please use brackets rather than superscript for reference numbers.

Comments from Reviewers:

Reviewer #1: The authors have improved the paper and responded to the peer review comments.

Reviewer #2: Thank you for the opportunity to re-review this important, well-written manuscript. The authors have been responsive to reviewer comments. I have no further concerns.

Reviewer #3: Alex McConnachie, Statistical Review

The authors have appropriately addressed all of my original points, and I have no further comments to make.

[LINK]

---

## [Editor Report · Decision Letter 3]

1 Apr 2021

Dear Dr Jeong, 

On behalf of my colleagues and the Academic Editor, Lars Ake Persson, I am pleased to inform you that we have agreed to publish your manuscript "Parenting Interventions to Promote Early Child Development in the First Three Years of Life: A Global Systematic Review and Meta-Analysis" (PMEDICINE-D-20-04650R3) in PLOS Medicine.

PRESS

Sincerely, 

Caitlin Moyer, Ph.D. 

Associate Editor 

PLOS Medicine